



# A portable, robust, stable and tunable calibration source for gas-phase nitrous acid (HONO)

Melodie Lao[1], Leigh R. Crilley[1], Leyla Salehpoor[1], Teles C. Furlani[1], Ilann Bourgeois[2,3], J. Andrew Neuman[2,3], Andrew W. Rollins[2], Patrick R. Veres[2], Rebecca A. Washenfelder[2], Caroline C. Womack[2,3], Cora J. Young[1], and Trevor C. VandenBoer[1, *]

[1] *Department of Chemistry, York University, Toronto, ON*

[2] *NOAA Chemical Sciences Laboratory, Boulder, CO*

[3] *Cooperative Institute for Research in Environmental Sciences, University of Colorado, Boulder, CO*

[*] *Communicating Author: tvandenb@yorku.ca*

## Abstract

Atmospheric HONO mixing ratios in indoor and outdoor environments span a range of less than a few parts per trillion by volume (pptv) up to tens of parts per billion by volume (ppbv) in combustion plumes. Previous HONO calibration sources have utilized proton transfer acid displacement from nitrite salts or solutions, with output that ranges from tens to thousands of ppbv. Instrument calibrations have thus required large dilution flows to obtain atmospherically relevant mixing ratios. Here we present a simple universal source to reach very low HONO calibration mixing ratios using a nitrite-coated reaction device with the addition of humid air and/or HCl from a permeation device. The calibration source developed in this work can generate HONO across the atmospherically relevant range and has high purity (90-99 %), reproducibility, and tunability. Mixing ratios at the tens of pptv level are easily reached with reasonable dilution flows. The calibration source can be assembled to start producing stable HONO mixing ratios (RSE ≤ 2 %) within two hours, with output concentrations varying ≤ 25 % following simulated transport or complete disassembly of the instrument, and ≤ 10 % under ideal conditions. The simplicity of this source makes it highly versatile for field and lab experiments. The platform facilitates a new level of accuracy in established instrumentation, as well as intercomparison studies to identify systematic HONO measurement bias and interferences.

## 1. Introduction

In the lower troposphere, the hydroxyl radical (OH) is the principal daytime gas-phase oxidant, and will react with volatile organic compounds (VOC) to form secondary pollutants such as ozone ($O_3$) and secondary organic aerosols (Spataro and Ianniello, 2014; Ye et al., 2018). Photolysis of nitrous acid (HONO) is a direct source of the hydroxyl radical (OH) (R1). Consequently, this can be a significant contributor to the integrated daily OH budget, ranging from 4-56 % in urban areas (Lee et al., 2013; Volkamer et al., 2010), up to 80% in semi-rural areas in the winter (Kim et al.,


2014), along with additional vertical and temporal variability (Crilley et al., 2016; Young et al., 2012; Zhang et al., 2009).

$$HONO_{(g)} + hv\ (\lambda < 405\ nm) \rightarrow OH_{(g)} + NO_{(g)} \tag{R1}$$

The reported daytime mixing ratios of ambient HONO outdoors can vary considerably for different environments, ranging from a few parts per trillion by volume (pptv) in the clean remote marine and Arctic boundary layers (Honrath et al., 2002; Kasibhatla et al., 2018; Reed et al., 2017) to 10 parts per billion by volume (ppbv) in polluted megacities such as Beijing (Tong et al., 2016; Zhang et al., 2019). Measurements within biomass burning plumes from forest fires have shown very
high HONO levels, often up to 60 ppbv (Chai et al., 2019; Neuman et al., 2016; Veres et al., 2010b). There is a growing body of evidence that HONO concentrations can be significant in indoor environments, with levels up to 50 ppbv reported from gas stove cooking emissions (Collins et al., 2018; Gligorovski, 2016; Gómez Alvarez et al., 2012; Liu et al., 2019; Young et al., 2019; Zhou et al., 2018). There are a number of atmospheric HONO sources that have been reported:
direct emissions (e.g. vehicles and biomass burning); gas-phase homogenous reaction of NO and OH, biological production in soils (Mushinski et al., 2019), and a number of heterogeneous surface reactions ((Spataro and Ianniello, 2014) and references therein). Despite the importance of HONO to atmospheric photochemistry and radical budgets, the contribution of these sources to observed HONO levels is still poorly constrained, particularly during the daytime (Gall et al., 2016;
Kleffmann, 2007; Lee et al., 2016; Oswald et al., 2013; Pusede et al., 2015; Sörgel et al., 2015; Tsai et al., 2018; Ye et al., 2016).

Due to the importance of HONO in our understanding of tropospheric photochemical oxidation and indoor atmospheric oxidation chemistry, accurate and precise quantitative measurements are required. However, gas-phase HONO has remained a challenging compound to measure due to
several instrument artefacts and interferences. Within inlet lines, positive artefacts can occur as a result of heterogenous HONO formation on wet surfaces (Kleffmann and Wiesen, 2008; Zhou et al., 2002), while the reactive nature of HONO can also lead to negative artefacts due to wall losses (Pinto et al., 2014). Furthermore, there can be interferences from ambient components in the atmospheric matrix, such as the reduction of $NO_2$ by numerous compounds, as well as particulate
nitrite (Kleffmann et al., 2006; Kleffmann and Wiesen, 2008; Sörgel et al., 2011; VandenBoer et al., 2014; Villena et al., 2011). Recent intercomparison studies have shown substantial differences between various HONO measurement techniques (Cheng et al., 2013; Crilley et al., 2019; Pinto et al., 2014; Stutz et al., 2010). Crilley et al. (2019) observed that while different HONO measurement techniques agreed on the temporal trends in HONO concentrations, the reported
absolute concentrations displayed systematic variation. Most studies are unable to pinpoint the exact cause of the observed divergence between instruments; it may be due to spatial heterogeneity in ambient HONO concentration, unknown chemical interference(s), and/or differences in the accuracy and precision of calibration approaches. A portable calibration unit compatible with all instruments/techniques could assist in ruling out systematic bias and identifying interferences
between instruments during intercomparison studies.



A variety of approaches have been used in the past to generate gaseous HONO standards. Most of these depend on acid displacement from a solution containing nitrite ($NO_2^-$) or from solid sodium nitrite ($NaNO_2$). Acids used have included sulfuric acid, hydrochloric acid, and oxalic acid, with evaporation of $NH_4NO_2$ also reported (Braman and de la Cantera, 1986; Febo et al., 1995; Taira and Kanda, 1990; Večeřa et al., 1991). By far the most widely employed modern HONO calibration methods stem from the report of Febo et al. (1995) who presented a system for generating a continuous source of stable gas-phase HONO in the tens of ppbv to parts per million by volume (ppmv) range. This system utilised the reaction between gas-phase hydrochloric acid (HCl) and $NaNO_2$ powder to generate gas-phase HONO, as described in R2:

$$NaNO_{2(s)} + HCl_{(g)} \rightarrow HONO_{(g)} + NaCl_{(s)} \tag{R2}$$

However, this calibration source requires a gas-tight vessel of HCl solution contained in a thermostatic bath that presents considerable difficulty for many field measurement applications. Adaptations include immersing thin-wall Teflon tubing in concentrated HCl, high concentration HCl cylinders, and HCl permeation devices. Gaseous HCl generated by these methods then mixes with loose $NaNO_2$ crystals in a stirred reactor (Stutz et al., 2000), dispersed using 3 mm glass beads packed in PFA tubing to increase porosity (Roberts et al., 2010), or pieces of PFA tubing (McGrath et al., 2019; VandenBoer et al., 2015; Zhou et al., 2018). These adapted approaches have been used to calibrate many atmospheric HONO instruments (Crilley et al., 2019; Heland et al., 2001; Ren et al., 2010; Roberts et al., 2010; Stutz et al., 2000; VandenBoer et al., 2013, 2015; Wang and Zhang, 2000; Young et al., 2012).

While widely used, the method described by Febo et al. (1995) presents several practical challenges. The typically high HONO mixing ratios generated by this approach (up to 20 ppmv) are challenging to dilute to atmospherically relevant mixing ratios. The high quantities also lead to auto-dissociation of HONO (R3), resulting in the production of nitrogen oxide impurities of NO and $NO_2$ (Febo et al., 1995; Neuman et al., 2016), and ClNO in the presence of HCl at ppmv levels (Gingerysty and Osthoff, 2020).

$$HONO_{(g)} + HONO_{(ads)} \rightarrow NO_{(g)} + NO_{2(g)} + H_2O \tag{R3}$$

Further, to reduce the variability in HONO output over time, the powdered $NaNO_2$ bed requires continuous mixing to maintain equilibrium, as well as a Teflon filter to prevent loss of $NaNO_2$ powder by entrainment in the gas flow. The degradation of the powdered $NaNO_2$ structure can limit the lifetime of the source and results in unstable HONO production rates (Febo et al., 1995; Gingerysty and Osthoff, 2020). Other systems using dispersed $NaNO_2$ suffer from sensitivity to vibration, causing changes in HONO output and limiting calibration accuracy (Zhou et al., 2018). Once operational, the original or modified methods require up to a day to stabilize and these systems must be kept continuously operating and stationary to preserve the HONO output stability.

One solution for producing gaseous HCl for acid displacement is to use a temperature-controlled permeation device (PD). A permeation oven is a simple instrument that can be used for the



preparation of low mixing ratios of gases from ppbv to ppmv levels (Veres et al., 2010a; Washenfelder et al., 2003). This approach has been used to generate a consistent quantity of gaseous analytes for over 400 compounds because it is low-cost, portable, and robust (Mitchell, 2000). Permeation devices are typically made of inert polymer tube of known permeability filled with a (semi-)volatile liquid. Both ends of the device are sealed either with caps or permeable plugs and the emission is determined by the surface area and thickness of permeable polymer, the concentration of the contained solution, and the temperature (O'Keeffe and Ortman, 1966; Susaya et al., 2012).

The aim of the current work was to make a portable and easy to assemble HONO calibration instrument compatible with HONO-measuring instruments commonly used within the atmospheric research community. We developed coated devices to facilitate reactions of sodium nitrite ($NaNO_2$) which release HONO when exposed to water vapour and HCl (R2). Herein we demonstrate that the $NaNO_2$-coated reaction devices produce a stable and continuous supply of high-purity gaseous HONO. The output of this HONO calibration source spans the range of environmentally relevant mixing ratios, up to tens of ppbv. The emission quantities, mass balance, and purity of gaseous HONO were determined through a series of control tests with various instruments. We present evidence of its robustness, reproducibility, and stability in HONO output. Finally, we evaluated methods to control the mixing ratio output of the calibration source and provide several approaches and recommendations on its use.

## 2. Experimental Methods

### 2.1 Coated $NaNO_2$ reaction devices

Reactions of $NaNO_2$ on humidified surfaces produce HONO. A large and consistent surface area is required to reproducibly produce HONO at the desired levels.

A $NaNO_2$ (EMSURE®; ACS Reag. Ph Eur, Germany) coating solution was made as a 20 g $L^{-1}$ $NaNO_2$ solution. The coating solution solvent was composed of equal volumes of methanol (HPLC Grade; Fisher Chemicals, Ottawa, ON) and 18.2 MΩ·cm deionised water with 1.0 g $L^{-1}$ glycerol (Sigma Chemical Company, St. Louis, MO, USA) to facilitate a uniform salt coating. The solution was made by dissolving the $NaNO_2$ in the water first, followed by the addition of the glycerol and then methanol. The coating solution was stored in an HDPE bottle wrapped in aluminum foil at 4 °C until needed and remade every three months. To coat a reaction device, 3 mL of coating solution was first transferred into a heat-straightened ½" (1.27 cm) PFA tube with a length of 14.4 cm and surface area of 86.2 $cm^2$. Rubber stoppers with centred 4.5 mm holes were inserted into both ends of the PFA tube to reduce solution loss while evaporating the solvent. The PFA reaction device was repeatedly inverted and rotated while covering both stopper holes to coat the inner surface completely. The reaction device was then dried by insertion into an 8" (20 cm) length of aluminum pipe (1-¼"/3.18 cm I.D.) and placed onto heated stainless steel rollers to evenly coat the PFA



150   reaction device surface as the solution evaporates over a few hours (Nostalgia Electrics, RHD800 Retro Series; or Great Northern Popcorn Company, 4078 GNP Hotdog 7 Roller Machine). Until their experimental use, prepared $NaNO_2$ PFA reaction devices were sealed with Parafilm or vinyl end caps (McMaster-Carr; P/N: 40005K14) and kept in a dark box at room temperature.

155   Teflon-coated aluminum annular denuders (URG-2000-30x150-3CSS, URG Corporation, Chapel Hill, NC) were also used in some experiments in place of the $NaNO_2$ device (Figure 1). To coat these denuders, 3.0 mL of the $NaNO_2$ coating solution was transferred to the device, followed by capping, inversion, rotation and shaking to ensure all concentric etched glass surfaces were coated. The excess $NaNO_2$ coating was decanted and the denuder dried with zero air at a flow of 1.0 standard litre per minute (SLPM) for about 10 min at room temperature.

## 160   2.2 Gas flows

The calibration source, which uses a permeation oven and $NaNO_2$ reaction device to generate HONO, was designed to be cost-effective, lightweight, and robust for use with dry compressed air as the carrier gas (Figure 1). Full technical details of the design rationale and assembly of the custom-built permeation oven can be found in the Supporting Information (SI, Sections S1-2, Figures S1-5), with only a brief description given below. A $NO_x$ analyzer was used to characterize the output from the HONO source. A single cylinder or zero air generator provided the separate carrier gas flows required for the PD, a humidifier, and a dilution flow.

Carrier gas flow through the permeation oven can be provided by a compressed cylinder of zero air or nitrogen (Praxair; Air Ultra Zero, 99.999%, AI 0.0UZ-K; High Purity Nitrogen, 99.998 %, NI 4.8, Toronto, ON) or in-situ zero air generator (e.g. Aadco Instruments Model 747-10, Cleves, OH) providing 20 psi of pressure to control the flow entering a four-way ¼" (64 mm) Swagelok cross fitting. The zero air flows through two critical orifices setting flows of ~50 sccm (Lenox Laser, Glen Arm, MD; SS-4-VCR-2-50) and a mass flow controller (MFC; MKS Instruments, Inc.; M100B00814CS1BV, 10 SLPM, gas; AIR, Kanata, Canada) set to deliver a dilution flow of 1.0 SLPM. A proportional-integral-differential (PID) temperature controller (Omega™; CN 7823, St-Eustache, QC) was used to regulate the temperature of a machined aluminum (Al) block. The first critical orifice connects to the HCl PD channel within the heated Al block and the second connects to a 25 mL glass impinger (EMD Millipore Corporation, Billerica, MA, USA) containing deionised water. The flows are combined and mixed to a relative humidity (RH) of 50 %, which then enters the coated $NaNO_2$ reaction device. The HCl drives the acid displacement reaction in the $NaNO_2$-coated PFA device, releasing HONO into the gas phase. The flow exits the oven into the dilution flow being delivered to an instrument or experimental system.



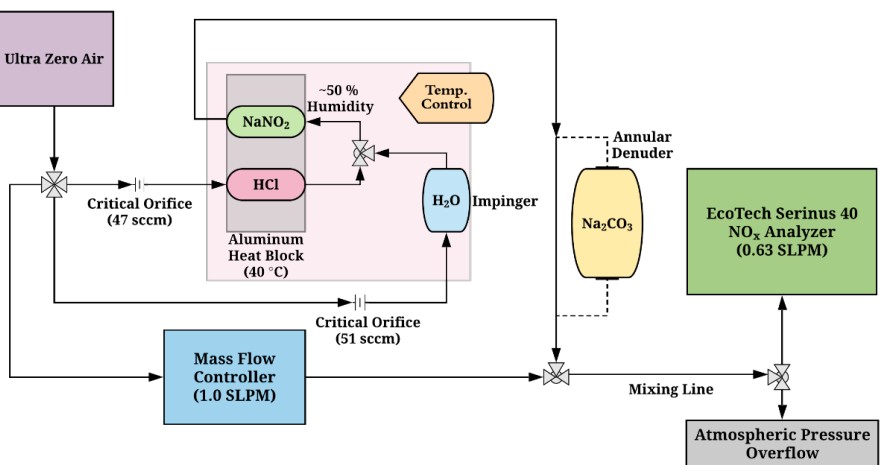

**Figure 1**. Flow and component schematic of the HONO calibration system (pink shaded region) interfaced with a $NO_x$ analyzer (green), dilution mass flow controller (blue), and an exchangeable $Na_2CO_3$ annular denuder (yellow). Lines with black arrows denote the direction of gas flow through system components. Tee and cross fittings are denoted by arrays of grey triangles.

## 2.3 Custom-built HCl permeation devices (PDs)

Although PDs are available from commercial suppliers, they are custom made here to reduce costs, as described in detail in the Supporting Information (Section S2, Figure S5). Briefly, custom PDs are made from PFA tubing (3mm ID with 5mm OD, P/N: 5733K73; McMaster-Carr, Aurora, OH) fitted with PTFE plugs (0.125"/32 mm diameter, P/N: 84935K64; McMaster-Carr). A series of HCl PDs were made as aqueous solutions to obtain PDs containing 1.2 M and 6 M HCl solutions (OmniTrace®; 34-37 %, HX0607-1, SigmaAldrich, Oakville, ON; Table 1).

## 2.4 $NO_x$ analyzer for HONO detection

The output from the HONO calibration source was monitored using a commercial chemiluminescent $NO_x$ analyzer fitted with a Mo catalytic converter, set to 325 ˚C (Serinus 40, American Ecotech, Warren, RI). The conversion efficiency of $NO_2$ to NO was calculated by combining known concentrations of NO from a standard cylinder (Praxair; NI NO5MC-A3, 4.88 (±5 %) ppmv, Toronto, ON) and $O_3$ using a gas calibration instrument (Gascal 1100TS, American Ecotech, Warren, RI). The conversion efficiency was determined according to the manufacturer specifications at 98.84 (±0.38 %) for $NO_2$ mixing ratios delivered to the system spanning 100 to 400 ppbv. While the Mo catalyst is meant to convert $NO_2$ to NO for detection by the analyzer, it is well known that HONO is also quantitatively converted to NO (Febo et al., 1995), and the conversion efficiency was determined experimentally (Section 2.6). A $NO_x$ analyzer was preferred to other independent calibration methods such as ion chromatography with conductivity detection



(IC-CD), as it is capable of continuous real-time measurement of HONO, allowing rapid frequent checks on the calibration source output stability.

During experiments, ~100 sccm from the HONO source was diluted into an additional 1.0 SLPM of zero air from which the $NO_x$ instrument sampled 0.63 SLPM (Fig. 1). The $NO_x$ analyzer measured NO on either the NO or $NO_x$ channels for an averaging period of 1 minute with the Kalman filter set to 60 s or 300 s. To correct for instrument drift or $NO_x$ contamination in the zero air, the analyzer was flushed for at least 15 min at the beginning and end of each experiment. An annular denuder coated with 20 g $L^{-1}$ sodium carbonate in 50:50 methanol and water solution ($Na_2CO_3$; ACS reagent >99.7%; SigmaAldrich, St. Louis, MO) – similar to that used here for $NaNO_2$ - was inserted during some experiments to scrub HONO from the experimental flow (Fig. 1). The denuder was prepared by transferring 10 mL of $Na_2CO_3$-coating solution, capping, then inverting and rotating to distribute the solution evenly. The remainder of the coating solution was decanted and the denuder surfaces dried under a flow of 1.0 SLPM of zero air until completely dry (~10 min). The denuder was inserted into experimental flows for at least one hour as a second check on sources of background NO and $NO_2$ as impurities being emitted from the calibration source or carrier gas. A $Na_2CO_3$ denuder can also be used as a robust alternative to provide the $NO_x$ analyzer inlet overflow instead of a cylinder of zero air. The enables corrections of HONO measurements or calibrations for $NO_x$ present in the sample air or calibration source carrier gas, respectively.

## 2.5 Conversion efficiency of the $NO_x$ analyzer Mo-catalyst for HONO

A Mo catalyst at 325 °C will reduce HONO to NO, though reports have shown that this conversion may vary between $NO_x$ analyzers (McGrath et al., 2019; Zhou et al., 2018). We measured the HONO generated by the calibration source with the $NO_x$ analyzer, then directed the HONO to a scrubbing solution of 1 mM NaOH in two glass impingers connected in series for several hours to days to collect $NO_2^-$ to a level that could be quantified by IC-CD. The second bubbler was used to determine the extent that HONO was quantitatively collected in the first bubbler (i.e. to capture any breakthrough). The HONO generated by the calibration source and quantified by IC was compared to the $NO_x$ analyzer measurement, using the introduction of a $Na_2CO_3$ annular denuder to perform background correction. The HONO conversion efficiency determined by comparison to the IC was found to be 104 ± 4 % (n = 3), confirming unit conversion efficiency, with the associated error set by the 4 % accuracy of the IC-CD method for $NO_2^-$ ($R^2$ > 0.999) when employing our previously developed separation method (Place et al., 2018). The IC precision near the analyzed concentrations for $NO_2^-$ was measured to be 3 %. All data presented in this manuscript therefore uses a conversion efficiency of unity for the Mo-catalyst.

## 2.6 Supporting Instrumentation

In our mass balance experiments for the production mechanisms governing HONO generation in the calibration system, we used two additional tools to monitor experimental gas flows. Mixing





ratios of HCl were measured at 0.5 Hz using a cavity ring down spectrometer (CRDS) (G2108
HCl Gas Concentration Analyzer, Picarro, Santa Clara, CA) with a 5 pptv detection limit for 1-
minute averaged data. Further details on the performance of this instrumentation can be found in
Dawe et al. (2019). Measurements of HCl and HONO to investigate acid displacement efficiency
of the calibration system were measured at 2 Hz using a quadrupole chemical ionisation mass
spectrometer (CIMS, THS Instruments LLC, Atlanta, GA) using acetate reagent ions to facilitate
proton transfer and detection of negative ions at m/z 35 and 46, respectively. Observed ions were
normalized to the detected quantity of the acetate reagent ion and multiplied by $8 \times 10^5$, resulting in
signal units of normalized counts, as we have previously reported for the detection of these analytes
(VandenBoer et al., 2013). Signal from the CIMS was averaged to a 1-minute time base to compare
to other measurements.

In our purity and stability experiments, additional instrumentation was used to detect HONO, $NO_y$,
and other reactive gases. A time-of-flight chemical ionisation mass spectrometer utilising iodide
adduct reagent ions ($I^-$ ToF-CIMS; Aerodyne Research Inc., Billerica, MA) was used to measure
HONO and detect a wide array of other analytes (e.g. $ClNO_2$, $HNO_3$, $N_2O_5$, etc.) in experimental
gas flows. Specific operational details of the $I^-$ ToF-CIMS for these atmospheric species are
presented elsewhere (Neuman et al., 2016; Veres et al., 2020). A broadband cavity enhanced
absorption spectrometer (ACES) was used to measure HONO and $NO_2$ (Min et al., 2016) and a
single-photon laser induced fluorescence (LIF) instrument was used to measure NO (Rollins et al.,
2020). A high sensitivity chemiluminescent NO instrument fitted with a gold catalyst ($NO_yO_3$)
was used to quantify NO and $NO_y$ (Fahey et al., 1985; Fontijn et al., 1970; Ridley and Grahek,
1990; Ridley and Howlett, 1974; Ryerson et al., 1999).

# 3 HONO calibration source characterization

## 3.1 NaNO₂-coated reaction device

Previous calibration methods required a 1-2 g bed of loose crystalline $NaNO_2$ to generate high
mixing ratios of HONO, but only consumed a minimal amount of $NaNO_2$ from the total supply
before being thrown away (Febo et al., 1995; Roberts et al., 2010). At maximum, our $NaNO_2$
coated PFA reaction devices could contain up to 60 mg of $NaNO_2$ (3.0 mL x 20 g $L^{-1}$ $NaNO_2$
coating solution) or 40 mg of $NO_2^-$ if there was 100 % coating efficiency. Due to the hydrophobic
nature of PFA and the loss of liquid solution from the drying procedure, however, the reaction
device retained only a fraction of the applied $NaNO_2$. The amount of $NO_2^-$ present after coating
the PFA devices (n = 3) was determined by rinsing with deionised water and analysis by IC-CD.
An average mass of 0.53±0.27 mg $NO_2^-$ was deposited on the surface of the PFA device, 1.3 % of
the total $NO_2^-$ applied. The quantity coated on the PFA devices was sufficient to generate stable,
low mixing ratios of HONO for extended periods. To efficiently use most of the salt, we calculated
how long the $NaNO_2$ coating could provide a specific calibration mixing ratio of HONO (E1).
Thus, we designed and operated our coated devices based on their calculated capacity to generate



a specific mixing ratio of HONO ($C_{HONO}$) continuously over time based on the number of moles of $NaNO_2$ deposited in the coating ($n_{NaNO2}$) and the total dilution flow in moles of air for that same duration ($F_{air}$).


$$C_{HONO} = n_{NaNO2} / F_{air} \qquad\qquad (E1)$$

To generate higher mixing ratios of HONO, more $NaNO_2$ mass and/or coated surface area are required. The higher surface area of a coated glass annular denuder housed in Teflon-coated aluminum tubing was explored for use as an alternative to PFA tubing. To test this, three annular denuders were prepared using the same volume of coating solution. An average mass of 7.26±1.80

mg of $NaNO_2$ on the denuder surface was determined, 18 % of the total applied. Thus, the coated annular denuder resulted in about eighteen times more deposited $NaNO_2$ than the PFA devices, due to the higher available surface area of pattern-etched glass. Unfortunately, these devices proved unstable, as discussed below, and are expensive. The HONO output from other tubing materials were also tested (Section 3.7.2).


The lifetime of the $NaNO_2$ devices can be approximated using E1, under the assumption that a stable output of HONO is generated from the start of the experiment. At standard room temperature and pressure a device generating 2 ppbv of HONO and containing the average 0.53 mg of $NO_2^-$ observed for the PFA device could last for up to 88 days. In practice, a PFA device generating 2

ppbv min$^{-1}$ of HONO was observed to last approximately a month during experiments performed to test the stability and reproducibility of the PFA devices (Sections 3.5, 3.6). The lifetime of the device is expected to decrease proportionally if a higher output of HONO for a given mass of $NaNO_2$ coating is required. Decreasing HONO mixing ratios on the order of a hundred pptv on hourly timescales (for an initial few ppbv of output) was used as a metric to indicate that coated

reaction devices were depleted since their output was no longer stable.

## 3.2 HONO generation with water vapour

Prior calibration sources have exclusively reported HONO production via the acid displacement mechanism. In the mass balance experiments reported below, where we employ this mechanism,

it was discovered that water vapour alone was responsible for a measurable amount of the generated HONO in the ppbv regime. Mixing ratios of HONO produced using our coated PFA reaction devices exposed to water vapour were too low to accurately measure using our $NO_x$ analyzer (≤ 0.6 ppbv). To explore the influence of water vapour (i.e. humid air) on HONO output, we performed a series of experiments at different RH using an $NaNO_2$ coated annular denuder.

The denuder generated higher HONO mixing ratios, on the order of several ppbv in 1.1 SLPM. Prior to the experiments, the calibration source unit was flushed with zero air for at least 12 hours. The absence of HCl (< 5 pptv) was confirmed with the CRDS. When the RH passing through the denuder was 0 % we observed no HONO, with signal near the detection limit of the $NO_x$ analyzer (0.50±0.48 ppbv, n = 43). When we increased the RH of the carrier gas, we observed the production

of HONO, but the variation was not monotonic. At a RH of 25 % HONO output increased to





11.73±0.39 ppbv (n = 35) followed by a decrease at an RH of 50 % to 8.60±0.63 ppbv (n = 38). This trend is likely due to the effective Henry's Law constant of HONO in the aqueous film on the surface of the NaNO$_2$ device, due to the weak acid nature of HONO (pK$_a$ = 3.4). When the humidity is higher, less HONO may be released from the surface due to the increased presence of water in which a larger equilibrium concentration of aqueous nitrite can be sustained. This contrasts with HCl (pK$_a$ of -8), which completely dissociates in aqueous solution on the surface of the NaNO$_2$ device and facilitates the acid displacement mechanism (R2).

This is the first observation of water vapour-produced HONO. Prior calibration sources typically generated very high HONO mixing ratios from 100 ppbv up to tens of ppmv in the displacement vessel, resulting in the contribution from humid air being undetected (Febo et al., 1995; Gingerysty and Osthoff, 2020; McGrath et al., 2019; Roberts et al., 2010; VandenBoer et al., 2015; Zhou et al., 2018). The observed HONO mixing ratios from this mechanism in our experiments would likely be within error of the mass balance calculations, or indistinguishable from noise in the analytical instrumentation in prior reports. Our results suggest that the use of water vapour passed over a NaNO$_2$ coated PFA reaction device produces sub-ppbv mixing ratios of HONO for calibration of instruments making ambient observations in remote environments. Using water vapour alone, the only way to increase the HONO mixing ratios from the calibration system is to increase the available amount of NaNO$_2$, which is challenging (Sections 3.1 and 3.7.2). A more controlled approach to reach higher mixing ratios is through the acid displacement technique.

## 3.3 HCl emissions from custom-built PDs

To generate stable HONO mixing ratios using an NaNO$_2$ reaction device on the order of a few to tens of ppbv, a stable source of HCl is required. The HCl generated from custom-made PDs was therefore evaluated as a function of solution concentration contained (1.2 – 6 M), temperature (30 – 60 °C), and stability by CRDS (Table 1, Figure S6). Custom-made PDs of different concentration and lengths were tested for their ability to produce a range of HONO mixing ratios. Custom-made PDs have been previously demonstrated in our work to provide a stable emission source of HCl (MacInnis et al., 2016). The HCl output was found to be temperature-dependent and increased exponentially with temperature, as expected from theory (Section S2). However, as the PD was ramped to higher temperatures (> 50 °C) the permeation rate became more unstable, with a resulting settling time of about an hour as the materials from the permeation oven apparatus re-equilibrated (Figure S7). Since the HCl PDs were observed to be most stable at 30 °C and 40 °C, these temperatures were considered optimal to generate the stable HONO mixing ratios. Note that the HONO mixing ratios in the 100 sccm flow exiting the reaction device range from 9.7 to 72 ppbv (Table 1), which are much lower than all prior calibration sources, enabling easy dilution to reach environmentally relevant HONO mixing ratios for instrument calibration or experimental applications.





**Table 1**. Description of custom-made HCl permeation devices used to generate HONO. Zero air-corrected mixing ratios of emitted HCl and generated HONO using a single NaNO$_2$-coated PFA reaction device were measured with the heated Al-block at 40 °C in 1.1. SLPM.  The variability reported for each observation represents one standard deviation from the mean (n = 30 to 60 using 1-minute averaged data).

| PD | HCl (M) | Date of Manufacture (YYYY/MM) | PFA Device (cm) | PTFE Plug (cm) | HCl (ppbv) | HONO (ppbv) | Measured Date (YYYY/MM) |
|---|---|---|---|---|---|---|---|
| PD-1 | 1.2 | 2017/04 | 9.92 | 0.60 | 0.58 ±0.01 | 0.95 ±0.51 | 2019/10 |
| PD-6a | 6 | 2017/04 | 9.41 | 0.70 | 0.21 ±0.01 | 0.88 ±0.4 | 2019/10 |
| PD-6b | 6 | 2019/04 | 9.11 | 0.75 | 2.0 ±0.01 | 2.8 ±0.41 | 2019/11 |
| PD-6c | 6 | 2019/04 | 9.65 | 0.70 | 5.0 ±0.3 | 6.2 ±0.5 | 2019/11 |


Two newly made 6 M HCl PDs (-6b and -6c) were found to emit different, yet highly stable (e.g. ±0.01 ppbv), mixing ratios at identical oven temperatures (Figure S6). This demonstrates potential variability with each new device due to inconsistent results during custom fabrication compared to commercial PDs. The most likely source of such differences in output is variability in our sealing

of the PTFE plugs resulting in increased emission rates. In any case, the PDs remain stable with less than 10 % relative standard deviation. In comparison, commercial device emission rates are often certified within ±30 %. The emission rates of commercial PDs are certified through measurement by gravimetric weight loss over time (ng min$^{-1}$). A commercial ¼" (64 mm) Teflon HCl PDs of 6.55 M certified to emit 1905 ± 520 ppbv in 100 sccm flow at 40 ˚C (RSD = 27.3 %;

VICI Metronics, Inc.; Poulsbo, WA), has this output variance due to the co-emission of water and propagated measurement uncertainties. A lower variance in the emitted HCl was observed from our custom-made PDs when we quantified HCl directly by either CRDS or IC-CD. Custom-built PDs were therefore chosen over commercial PDs due to their demonstrated stability and low cost. It was found that HCl outputs of the custom-PDs slowly diminished over time, which emphasizes

the need for regular calibrations. For example, the HCl output from two-year-old PD-6a emitted 0.21±0.01 ppbv in 1.1 SLPM in comparison to 2.0±0.01 ppbv when it was newly made, which decreased the resulting HONO generation in the reaction device. Similar results have been observed in calibrations with PDs of aqueous NH$_3$ and HNO$_3$ solutions decreasing by ~30 % during two years of storage, as well as for carbonyl sulfide (Fried et al., 1998; Neuman et al.,

2003). Despite the decreasing HCl output over a year or more of use, HCl PDs act as a stable acid source on the order of weeks, producing consistent HCl output to subsequently generate stable HONO, even when removed from the permeation oven or stored for up to two months. Overall, it is difficult to replicate PD emission rates using the same HCl concentration and material dimensions for a custom-PD. The custom-built PD seals can be altered by replacing the PTFE plug

by crimping the ends of heated-to-pliability PFA tubing to form welded polymer ends (Section S2). Such an approach is expected to improve the reproducibility of the custom-device emission rates but is beyond the scope of this work to explore in more detail.



### 3.4 Acid displacement to generate HONO

Two techniques were used to assess the reaction completion between HCl and NaNO$_2$ in the calibration system. We applied a mass balance approach that combined the CRDS measurement of HCl, our NO$_x$ analyzer HONO measurement, and IC-CD quantitation of these acids scrubbed into 1 mM NaOH. The displacement efficiency was further confirmed by simultaneous observation of HCl and HONO by acetate quadrupole CIMS.

### 3.4.1 Mass balance of HONO generated

Experiments were conducted to confirm that HONO can be generated by introducing only humid air (50 % RH) within the NaNO$_2$ devices without the presence of HCl. In humid air, we observed HONO levels above the detection limit (DL) of the NO$_x$ analyzer. A single PFA device exposed to humid air (50 % RH) released up to 0.61 ppbv of HONO – equivalent to 77 % of the total HONO generated when coupled with an HCl PD (Table 2). The reaction of the humidified NaNO$_2$-coated

device, resulting in the release of HONO, implies formation of NaOH. Further speculation on the reaction mechanism is beyond the scope of this work. Given the existing challenge in producing low mixing ratios of HONO in the pptv range, it appears that these can be reached most easily without the use of an HCl PD in our calibration system, while higher mixing ratios necessitate the addition of HCl (Section 3.7). The NO$_x$ analyzer signal was indistinguishable from zero when the

NaNO$_2$ reaction device was absent, but all other conditions were matched. This demonstrates that HONO was generated only within the NaNO$_2$ reaction device.

The total flow for all mass balance experiments was 1.1 SLPM (Fig. 1) with zero air flows replacing those typically carrying reagents when they were removed. We observed that the HONO output from the reaction devices was greater than the HCl input from the PDs, confirming that

another chemical reaction was generating the remaining HONO (Table 2). Mass balance could only be achieved when accounting for the HONO generated by the NaNO$_2$ exposed to humid air (~50 % RH). No other acidic or ionic contaminants were present in NaNO$_2$ reaction devices or the HCl PDs when scrubbed solutions were analyzed by IC-CD. Therefore, other NO$_y$ species that could have biased the NO$_x$ analyzer measurement high were judged to be absent and pure HONO

was generated (i.e. only NO$_2^-$ was enhanced in calibration system flows scrubbed into 1 mM NaOH). Further investigation of the system HONO purity is presented in Section 3.7.3 which further supports this conclusion. The remainder of the HONO output from NaNO$_2$ devices quantitatively matched the HCl input to the reaction device in dry air after accounting for the water vapour production route. No HCl was observed to exit the devices, indicating unit acid

displacement efficiency and reaching mass balance.



**Table 2.** Mass balance of measured mixing ratios of HCl entering and HONO exiting the calibration source to determine acid displacement efficiency (ADE) at 50 % RH and 40 °C. Uncertainties represent 1σ standard deviation from the mean for ≥30 min of measurements and 1σ propagated error for calculated values.

| PD | $HCl_{IN}$ (ppbv) | HONO from $HCl+H_2O$ (ppbv) | HONO from $H_2O$ (ppbv) | HONO from $H_2O$ (%) | ADE (%) |
|---|---|---|---|---|---|
| PD-6a | 0.08 ±0.002 | 0.31 ±0.15 | 0.24 ±0.14 | 77 ±45 | >99 |
| PD-6b | 2.2 ±0.011 | 2.78 ±0.41 | 0.61 ±0.44 | 22 ±16 | >99 |

### 3.4.2 CIMS Measurements

Confirmation of these observations with the quadrupole CIMS provided higher time resolution observations of HONO and HCl simultaneously. The ions monitored were m/z 35 ($Cl^-$) for HCl and 46 ($NO_2^-$) for HONO (Figure 2). The instrumental sensitivity to these two analytes is similar under this ionisation scheme (VandenBoer et al., 2013). The HONO calibration source was stabilized for 2 h before the gas stream was introduced to the CIMS. Zero measurements were taken for 15 min before and after the measurements to correct for background drift in the m/z 46 signal. Upon sampling the output of the HONO calibration source the signal at m/z 46 rapidly increased (Fig. 2). The signal of $Cl^-$ at m/z 35 remained constant near zero throughout this period, confirming again that the HCl from the PD was entirely consumed by the $NaNO_2$ reaction device throughout the measurement period, consistent with the experiments presented above where no HCl was measured by the CRDS. Overall, the results from these assessments indicate that the HONO calibration source is generating HONO with a one to one displacement efficiency by HCl, consistent with this observation from other HONO calibration sources using higher quantities of HCl in a salt bed (Febo et al., 1995; Roberts et al., 2010), and the remainder originating from the water vapour reaction.

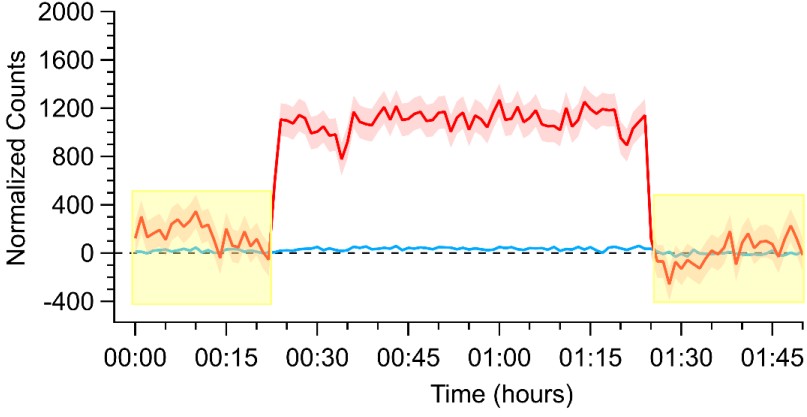



**Figure 2.** Conversion efficiency of HCl (blue) to HONO (red) via the acid displacement reaction on a NaNO$_2$ reaction device. The HCl PD-6a and one coated PFA device were used and measured following two hours of stabilization. The acids were observed by acetate quadrupole CIMS with time resolution of 0.50 s and averaged to 60 s. Yellow shaded regions indicate the addition of zero air to the instrument inlet for background correction, while red and blue shaded regions correspond
to 1σ variance in the observations.

## 3.5 Stability of HONO production

The time required to achieve stable HONO signals was tested by inserting HCl PD-6a and new NaNO$_2$ PFA reaction device into the calibration system, followed by flow start up. Stable HONO signals were observed within 7 h of powering on the HONO calibration system. This is 5 h longer
than required to reach stable mixing ratios for a previously stabilized NaNO$_2$ device. Three trials using newly coated NaNO$_2$ reaction devices and PD-6a, once stabilized, generated an average HONO output of 2.28±0.58 ppbv, which corresponds to an RSD of 24 % between runs and an RSE of 3% (n = 2367; Figure 3). The noise observed in the stabilized HONO output in Figure 3 can be primarily attributed to the noise associated with the NO$_x$ analyzer detector (18 of the 24 %; DL =
0.4 ppbv; 1-minute average). This conclusion is supported by the lower noise in ~2.5 ppbv HONO mixing ratios observed by the CIMS (Fig. 2, RSD of 8.1%), ACES (RSD 8.2 %), and NO$_y$O$_3$ (RSD 1.9 %). In these added observations with higher sensitivity instrumentation, the stability was equal to instrumental precision. This represents a major improvement over our previously reported calibration sources with potential for 30 % variability at a minimum (VandenBoer et al., 2013;
Zhou et al., 2018).

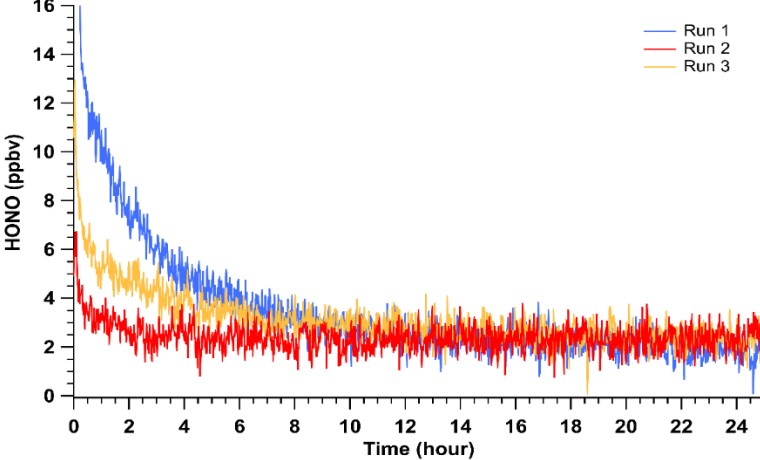

**Figure 3**. Mixing ratios of HONO observed using HCl PD-6a and three different, but freshly coated, NaNO$_2$ PFA reaction devices. Time zero indicates the start of HONO production in the calibration unit where no prior flow through the calibration unit existed, but all temperatures were
stable at 40 °C. Reported measurements are one-minute average data with a 30 s Kalman filter on the NO$_x$ analyzer.

However, when the HCl output from PDs is unstable, this can interfere with the stability of the HONO generated because it is dependent on acid displacement. A common characteristic of our custom-PDs monitored by real-time CRDS measurements are short-duration increases in output
over min, up to 1 h, due to reduced emission of $H_2O$ and increased emission of HCl, resulting in transient pulses from the device (Figure 4a). The anticorrelation between HCl and $H_2O$ is expected for a constant mass emission to result from the contained aqueous solution. A corresponding rapid increase in HONO production results from such occurrences (Figure 4b).

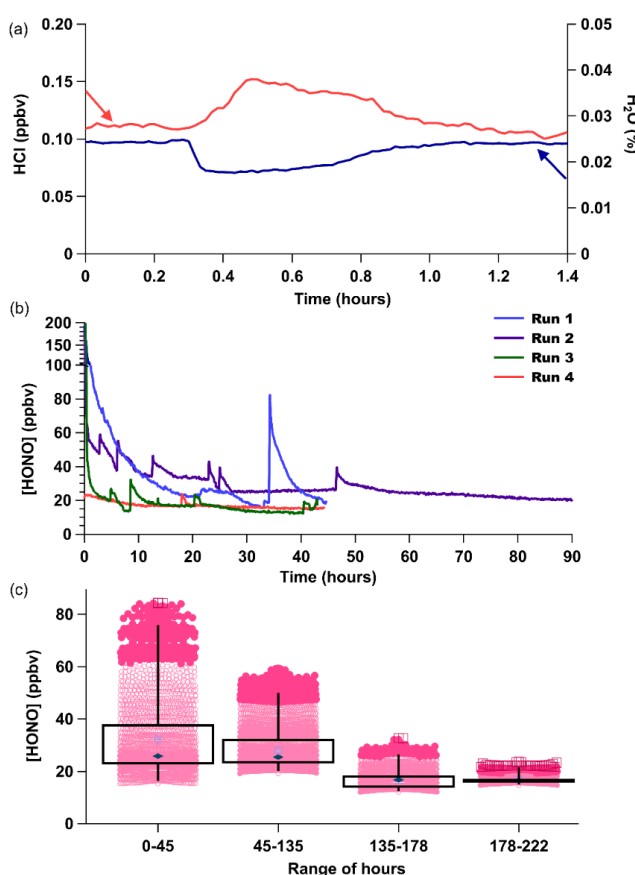

**Figure 4**. (a) CRDS high-resolution observation an HCl emission pulse (red) and $H_2O$ decrease (blue) from PD-6a resulting in 50 % increase of the HCl mixing ratio emitted. (b) Time series of four consecutive measurement periods of HONO production, using only PD-6c and a new $NaNO_2$ coated reaction device in each run. (c) Box (1st and 3rd quartile) and whiskers ($3\sigma$ from the mean) of HONO mixing ratios observed for the four runs are binned by duration of use for each new
reaction device in hours. Mean values are indicated with a filled dark blue diamond marker, median values by the light blue crossed box marker, dark pink circles are $2\sigma$ outliers and dark red squares $3\sigma$ outliers



Commercial PD manufacturers evaluate average mass emission rates by gravimetric weight loss over several weeks at 40 °C for certification, which could include such short-term events. The
HONO output from a newly made custom-HCl PD (PD-6c) over four consecutive observation periods upon insertion of a new $NaNO_2$ reaction device (Runs $1 - 4$) at constant temperature (40 °C) show that the new custom-PD requires about 1 week of operation before its output is stable (Figure 4b-c). Therefore, careful preparation of PDs and $NaNO_2$ reaction devices in advance of extensive use will yield a HONO calibration source with the fastest stabilization times possible for
continuous operation over a period of months. Note again, that the HONO measurements for Figures 5b-c were performed several months before the HCl emission rate for PD-6c presented in Table 1 were obtained, resulting in high HONO mixing ratios produced in these experiments.

## 3.6 Reproducibility and robustness

The HONO calibration system was designed to not only be stable, but reproducible in its output of HONO for a given PD and any $NaNO_2$ reaction device, resulting in robust portability. We tested the reproducibility, and therefore robustness, of the HONO calibration system by putting it through a series of experiments designed to simulate transport to, and use in, the field. Further assessment of its reproducibility by measuring the output with different $NaNO_2$ reaction devices and HCl PDs
were also made.

### 3.6.1 Field transport simulations

Simulations of field transportation subjected the system to full disassembly and reassembly of the acid displacement and permeation oven setup. In addition, for some experiments the calibration unit was transported on a lab cart over very rough flooring to simulate vibrations experienced for
real use when transported using rolling carts, mobile labs, or aircraft. For the first eight simulations PD-6a and one $NaNO_2$ coated reaction device were used over several weeks (see Table S1 for further detail). Following reassembly after the field transport simulations, the HONO calibration source was restarted, the system was equilibrated for 2 hours, and then its output measured by the $NO_x$ analyzer (Figure 5). An $Na_2CO_3$ coated annular denuder was incorporated into the middle of
five of the eight trial experiments for an hour to determine whether any $NO_x$ was being generated between restarts and its associated variability (FS1-FS5; Table S1). No measurable $NO_x$ was detected in any of these experiments.

The average HONO mixing ratio within the eight field transport simulations (FS) ranged from 1.68 to 2.51 ppbv. The HONO output across all eight field simulations had an average of 2.07±0.48
ppbv (RSD = 24 %; RSE = 2 % (n = 218)). These HONO mixing ratios are similar to the average HONO output of 2.28±0.58 ppbv (RSD of 24 %; RSE of 3 % (n = 2367) from the previous measurements with PD-6a (Figure 3), which were not subject to field simulations but did use freshly coated $NaNO_2$ reaction devices. The generated HONO mixing ratios varied most between our early experiments (FS1-FS4; RSD ≥ 24 %), when first gaining experience in ensuring gas-
tight connections throughout the calibration system, with improvement clearly emerging over time



(FS4-FS8; RSD ≤ 10 %). The RSE values of field transport simulations had a lower RSE of 1 % compared to 3 % for the experiments that were stationary (Fig. 5), likely due to the reuse of the same $NaNO_2$ reaction device. This demonstrates that the HONO calibration source can robustly generate a reproducible mixing ratio output within 25 % of the mean during each system

reconstruction if the same HCl PD is used. It is worth noting here again that most of the variance observed in HONO mixing ratio output within any of the presented trials derives from the precision of our $NO_x$ analyzer detector (Section 3.5).

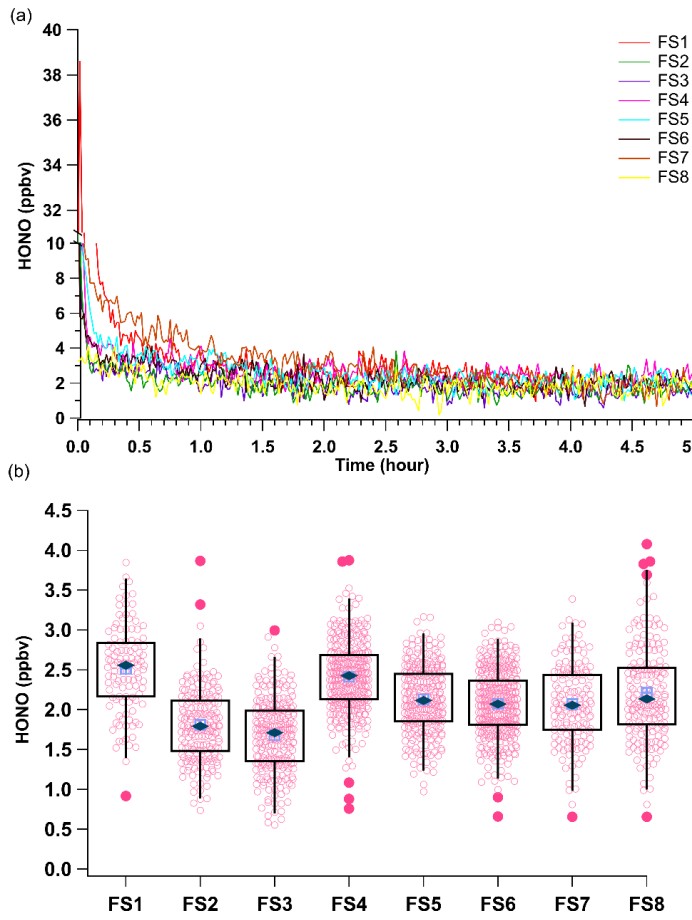

**Figure 5.** (a) Mixing ratios of HONO for the eight field transport simulations (FS, Table S1). All
observations were background corrected by linear interpolation across the experiments using zero air before and after HONO observations and an $Na_2CO_3$ coated annular during (Fig. S4). (b) Box and whiskers plot of the HONO output using measurements collected after two hours of calibration source stabilization. The light blue crossed box represents the median, the dark blue diamond the mean, light pink circles the data points, dark pink circles the 3σ outliers, and the black box the 1st
and 3rd quartiles of observed HONO mixing ratios. The whiskers denote the 3σ standard deviation.



### 3.6.2 Factors affecting reproducibility of HCl input

As shown in Table 1, PDs made with the same HCl concentration (6 M) and similar dimensions did not lead to the same HONO output, due to variability in the HCl emission rates. While it is possible for custom-made PDs to have similar HCl emissions and therefore HONO output (when using the same $NaNO_2$ device), it is difficult to achieve in practice. When making a new PD as per the methods described in Section S2, it can be difficult to replicate because the emission rate depends on the effectiveness of the plug seal. For this reason, one cannot simply make a HCl PD with plugs using the same concentration and material dimensions and necessarily expect the same output. We present an alternative Teflon welding method to overcome this limitation in Section S2 which has been successfully used for generation of VOC PDs. Regardless, the output of new HCl PDs should be quantified prior to use and not subject to extreme conditions to ensure the polymer permeability is retained.

The reproducibility of HONO output using a stable HCl PD is shown in Figure 5. The observed HONO output ranged from 1.68-2.51 ppbv (n=8, RSD = 24 %). We next tested the reproducibility for newly made HCl PDs. Two experiments used PD-6b, containing 6 M HCl (Figure S10). After a period of stabilization, the two experiments generated similar HONO mixing ratios (2.58±0.43 ppbv after 25 hrs, RSD = 16.5 %, RSE = 1.43 %, n = 792). The spikes in HONO output at 15 h and 21 h in the second experiment (green trace, Fig. S10) were likely due to pulses of HCl which we commonly observed with new PDs (e.g. Fig. 4a-b). This emphasizes our recommendation that new custom-HCl PDs should be used for an extended period prior to use for acid displacement to ensure the emission rate has stabilized.

We made another PD with 1.2 M HCl, as it emits less HCl in comparison to a PD made with 6 M HCl (Table 1), to determine the reproducibility in HONO output at lower mixing ratios. Across three experiments using a previously stabilized $NaNO_2$ device, an output of 0.69-1.12 ppbv (RSD = 53.7 %, RSE = 4.52 %, n = 143) was observed (Fig. S11). The high RSD is due to instrument noise as the HONO output approached the detection limit of the $NO_x$ analyzer (0.4 ppbv). Nonetheless, a stable output of HONO was achieved within two hours after starting the calibration system, similar to our previous results (Fig. 5). As long as a new custom-HCl PD has been allowed adequate time to stabilize under a gas flow at constant temperature (ideally 7 days), a stable HONO output can be easily replicated within two hours of starting the resulting HONO calibration system. We recommend quantifying the HCl emissions prior to use if the PD has been stored for a long period or been subjected to extreme conditions.

### 3.7 Adjusting and controlling HONO mixing ratios

Increasing the mixing ratio of HCl, and the type and quantity of $NaNO_2$ reaction devices connected in series were explored as methods to adjust the HONO mixing ratio exiting the calibration system.





### 3.7.1 Temperature control

The HONO calibration system was designed to be tunable by adjusting the oven temperature. HCl emissions increased with increasing temperatures (30 ˚C – 60 ˚C, Fig. S6), The HONO mixing ratios increased exponentially with increasing oven temperatures (Table 3 and Fig. S12). Very low levels of HCl exited the $NaNO_2$ device ($\leq$ 3% of HCl input), which demonstrated that there was continued near-unity acid displacement efficiency. With increasing temperature of the $NaNO_2$ reaction device in the presence of water vapour, a similar increase in HONO mixing ratio was observed, roughly doubling for every increase of 10 °C. Thus, the HONO mixing ratio output can be adjusted by changing the temperature of the Al-block with either water vapour alone or in combination with an HCl PD. The observed higher variability in HONO emissions at 50 ˚C was likely due to unstable emissions of HCl at this temperature (e.g. see Fig. S6). The use of multiple HCl permeation tubes in a single oven, in series, or in parallel are additional options to control the HONO mixing ratio generated in the calibration system.

**Table 3.** Average mixing ratios of HCl input (PD-6a), and HONO emitted from reaction with water vapour and with both reagents as function of temperature. Uncertainty denotes 1σ standard deviation from the mean of measured values.

| Temperature (°C) | HCl (ppbv) | HONO from $H_2O$ (ppbv) | Total HONO (ppbv) |
|---|---|---|---|
| 30 | 0.230 ±0.003 | 0.3 | 0.5 ±0.4 |
| 40 | 0.330 ±0.007 | 0.7 | 1.0 ±0.5 |
| 50 | 0.660 ±0.037 | 1.3 | 2.0 ±0.5 |

### 3.7.2 HONO output with different types of NaNO₂-coated devices

The produced HONO mixing ratios were tested using different materials coated with $NaNO_2$ via the same methodology as the PFA devices (Section 2.1) to see if there was an improvement in output stability or increased emissions of HONO. The materials used were all cylindrical tubing with ½" (1.27 cm) inner diameters and were of similar lengths and surface area. The different materials that were tested showed similar HONO outputs (within variability), except for quartz (Table 4). The quartz tubing gave a notably lower HONO output compared to other materials. This may have been due to a poor coating efficiency on the surface. That we observed similar HONO outputs for the other materials could be due to the devices having the same internal surface area coated with $NaNO_2$, implying that HONO output is proportional to surface-available $NaNO_2$. The inside of a PFA device was etched manually every few mm in concentric circles in an attempt to increase the surface availability of $NaNO_2$, but no change HONO output was observed compared to the unetched device.





**Table 4.** Average measured HONO mixing ratios (ppbv) using different ½" inner diameter tubing. All results at 40 ˚C and using same HCl PD (PD-6c). Variability shown is 1σ from the mean.

| Device material | HONO (ppbv) |
|---|---|
| PFA | 6.20 ±0.50 |
| Etched PFA | 5.68 ±0.71 |
| Stainless Steel | 6.75 ±0.83 |
| Nylon | 6.06 ±0.61 |
| Quartz | 3.72 ±0.55 |

Two additional methods were tested to increase the available surface area in the $NaNO_2$ device: increasing the number of coated PFA reaction devices in series and using an annular denuder. The HONO output with two PFA devices connected in series increased when using either 2.0 ppbv (PD-6b) or 5.0 ppbv of HCl (PD-6c) at 40 ˚C and 50 % RH. We did not observe HCl breakthrough at the exit of the first PFA device, indicating that the increased HONO mixing ratio is the result of the water vapour reaction. We observed variability in the amount of HONO produced between the four PFA devices, ranging from 0.8 to 1.3 ppbv per device.

More HONO can be generated using the same PDs in conjunction with an annular denuder, which has a larger internal surface area of 3063 $cm^2$ compared to 388 $cm^2$ for the PFA device. HONO emissions using PD-6c and an annular denuder produced a factor of four higher mixing ratio 24.5 ±1.0 ppbv compared to 6.2 ±0.5 ppbv with a single PFA device, but it required 45 hours to stabilize. Again, the increase is due to promotion of the water vapour reaction. The major drawback of using an annular denuder is that the output drifted to lower mixing ratios continuously at a rate of a few ppbv per hour, which was not a feature of the PFA devices (Figure 6). The HONO output over any 4 hour period was reasonably stable (within 0.5 ppbv) following the first 24 hours of stabilization time, which suggests that a $NaNO_2$ coated annular denuder could be viable for short duration HONO calibrations if a secondary quantitative method is available to confirm its output (e.g. a $NO_x$ analyzer with a quantified HONO conversion efficiency). Overall, using a $NaNO_2$ coated annular denuder can provide higher HONO outputs than using PFA devices but requires at least daily independent verification.



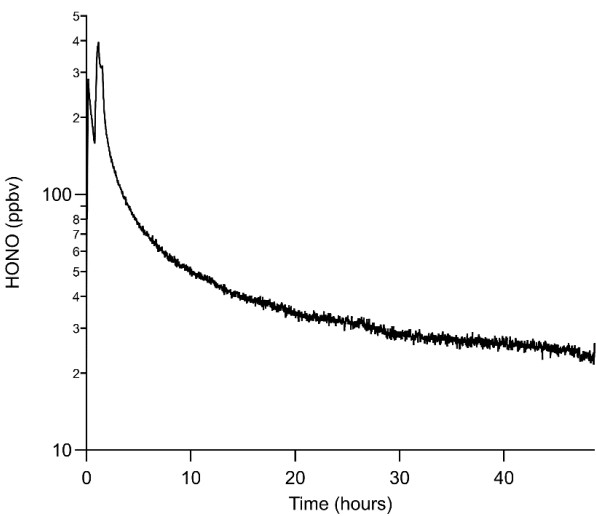

**Figure 6.** Mixing ratio of HONO produced from a NaNO$_2$ coated annular denuder using PD-6c. Time zero is when HCl was first introduced to the annular denuder at 50 % RH. Note that HONO mixing ratios are on a log scale.

### 3.7.3 Purity of the HONO output

Previous work has demonstrated that there can be a notable NO$_x$ impurity when generating HONO via the acid displacement method (Febo et al., 1995). To test the purity of the calibration source, the HONO output was analysed by additional reactive nitrogen, NO$_x$, and NO$_y$ instrumentation. For these experiments, we used a 6 M HCl PD and two PFA devices in series in a 40 °C calibration system, which was determined to have an output of 770 pptv of HONO. First, the output was analysed by an I$^-$ ToF CIMS and found no evidence for any detectable amounts of other nitrogen containing species (e.g. ClNO$_2$, ClNO, HNO$_3$; Figure S13) except for HONO (Neuman et al., 2016; Veres et al., 2020). The I$^-$ TOF CIMS is not sensitive to NO or NO$_2$, so further measurements were made with our Mo-catalyst NO$_x$ analyzer, a cavity-enhanced absorption spectrometer (Min et al., 2016), and a gold-catalyst NO$_y$ instrument (Fahey et al., 1985; Fontijn et al., 1970; Ridley and Grahek, 1990; Ridley and Howlett, 1974; Ryerson et al., 1999), which determined that NO$_2$ impurities were at or below 10 % of the generated HONO based on the detection precisions of the latter two instruments. Finally, we quantified NO impurities using a single-photon LIF instrument, which is sensitive to sub-pptv levels of NO (Rollins et al., 2020). We observed NO at 5.5 % of the measured HONO signal (42 versus 770 pptv). Examination of our modified NO$_x$ analyzer experiments using the same calibration system configuration revealed 6 % NO on average compared to the observed HONO (ca. 6-9 ppbv), consistent with the LIF measurements. In contrast, when using a NaNO$_2$-coated annular denuder with the same HCl PD, our modified NO$_x$ analyzer observed NO/HONO to decrease to 2%.





Recent work, using an analogous HONO calibration system, has found high production of NO, $NO_2$ and ClNO (> 10 %) when HCl input to loose $NaNO_2$ was > 4 ppmv (Gingerysty and Osthoff, 2020). We observed similarly high output of NO when the HCl input was increased to 2.4 ppmv

through the $NaNO_2$-coated devices. Under these conditions, the impurity may be due to self-reactions of HONO at high mixing ratios, as seen in other packed or stirred $NaNO_2$ salt beds (Febo et al., 1995). NO impurities at HONO mixing ratios below 100 ppbv in the salt bed may result from other heterogeneous processes generating NO in the lower HONO production regime. It may be that such small absolute quantities of NO have been produced in all prior calibration sources,

but as the mixing ratio of HONO produced has been reduced in our calibration system, that this impurity increases in a relative sense. The purity of the calibration source when generating < 100 ppbv in the salt bed is ≥ 90 % HONO, with the remainder accounted for as NO and/or $NO_2$.

### 3.8 Context and application

The RSD of our stable HONO output is < 2.5% and less than previous HCl acid displacement calibration source adaptations (VandenBoer et al., 2013; Zhou et al., 2018). Potential reasons for the improved stability in HONO output are the stable production of HCl from custom-PDs and that the calibration system presented in the current work eliminated the need for solid $NaNO_2$ powder, which is prone to disturbance of equilibrated emissions through vibrations that can result

in changes up to a factor of two in mixing ratio output (VandenBoer et al., 2013; Zhou et al., 2018). The RSD at the low HONO mixing ratios in this work are larger than reported by Febo et al. (1995), who generated much larger mixing ratios, but did not specify the measurement details of their $NO_x$ analyzer to facilitate true comparison. The greatest accuracy possible for this calibration source requires quantitation of the HONO output by a separate analytical method (e.g. IC-CD) and

should not rely on the assumption that the HONO generated is equivalent to the HCl delivered into the reaction device due to the additional production mechanism driven by water vapour. While the output of this system is demonstrated to be highly reproducible with a given HCl PD, we recommend regular calibration.

### 4 Conclusions


We present a cost-effective, portable, stable, tunable, and robust gas-phase HONO calibration source. We utilised both a water vapour only, as well as its combination with the acid displacement reaction of HCl, with sodium nitrite ($NaNO_2$) coated on the inner wall of a short length of PFA tubing within a machined Al-block permeation oven to produce a stable and continuous supply of

high purity gaseous HONO. We demonstrated for the first time that HONO was produced by humid air in the $NaNO_2$ reaction device, such that the HONO output was consistently higher than the HCl input. If a HONO calibration source in the pptv range was desired, it could be achieved easily by using only humid air flowing through an $NaNO_2$ coated reaction device. The output of this HONO calibration source spans the range of environmentally relevant mixing ratios - from



pptv levels to tens of ppbv. This will allow instruments to be calibrated and/or intercompared using their standard atmospheric sampling parameters, without the need for excessive - or impossible – dilution; nor additional pumps, valves, and mass flow controllers.

We demonstrated that our HONO calibration system mixing ratio was tunable by adjusting the temperature of the permeation oven to control the water vapour reaction, as well as HCl emission
rates from PDs. The most stable HONO output was achieved using $NaNO_2$ coated PFA devices at 40 °C, with HONO mixing ratios of 2.28±0.58 ppbv (RSD of 24 % and RSE of 3%, n = 8) that were reliably reproduced following complete assembly of the system. From our wide range of instrumental observations, the output of the source appears to be constant within ±10 % or better. The resulting system can be disassembled, transported, and reassembled to produce the same
HONO mixing ratios reproducibly, without the need for regular maintenance – where the same PD is retained between rebuilds. While higher HONO outputs were possible to generate using an $NaNO_2$ coated annular denuder for any given HCl PD, the outputs were unstable over time.

This HONO calibration instrument provides a universal solution to gas-phase HONO calibrations suitable for the full range of atmospheric instrumentation used for outdoor or indoor field
measurements or laboratory experiments. This calibration unit could be used to intercompare the responses/measurements between HONO instruments to investigate and validate accuracy and precision of their ambient measurements in addition to identifying and isolating potential interferences (Crilley et al., 2019). We anticipate it will also find utility in the generation of isotopically-labelled HONO for the emerging exploration of stable-isotopic composition of HONO
and its relation to the wide variety of suspected atmospheric HONO sources (Chai et al., 2019).

## Author Contributions

TV, CY, and RW conceptualized the calibration source. TV and CY supervised the experiments, acquired the funding, and provided the resources to support this work. TV designed the experiments, guided the investigations, and managed the project. ML constructed the custom PDs
under the guidance of TF and LS. ML and LS built the custom permeation oven and analyzed IC-CD samples. LC and ML performed the mass balance, reaction device, and purity experiments. LC and ML established and validated the methodologies. IB, AN, DR, PV, RW and CW provided instrumental resources, and performed measurements included in the stability and purity experiments. LS created the schematics and wrote the detailed description of the custom
permeation oven. ML constructed the reaction devices, performed part - or the entirety - of the experiments, and prepared the manuscript with contributions from all authors. All authors participated in data analysis, and the review and editing of the manuscript.

*The authors declare that they have no conflict of interest.*

## Acknowledgments

The Authors thank E Gaona-Colman for help in collecting the CIMS data, as well as J Liggio and J Wentzell for enabling the use of and training on the quadrupole CIMS. ML acknowledges



research support from an NSERC Undergraduate Student Research Award, and travel support through a York University Fieldwork Cost Fund award, respectively. LS acknowledges research support from the Harold I Schiff graduate award in Atmospheric Chemistry. LC and TV acknowledge travel support from the York University Faculty of Science Junior Faculty Fund. CY acknowledges support for this project through an NSERC Discovery Grant. TV and CY acknowledge funding for the instrumentation developed and used in this work provided by the Alfred P. Sloan Foundation Chemistry of Indoor Environments program (G-2018-11051).


## Data Availability


Datasets are presented in figures and summarized in tables throughout the main manuscript and supporting information. Raw data from these resources are available from the corresponding author upon request.

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
