# Peer review of "A portable, robust, stable and tunable calibration source for gas-phase nitrous acid (HONO)"

_Atmospheric Measurement Techniques, 2020_

## Referee Comment (RC1) · Anonymous Referee #1 · 8 Jul 2020

In the manuscript by Lao et al. the development and characterisation of a portable calibration source of HONO is described. The generation of HONO is based on a former source developed by Febo et al. (1995) and which is based on a reaction of gaseous humidified HCl with solid sodium nitrite. In contrast to the former study HCl is produced by a modified custom-built permeation source and the sodium nitrite is simply deposited on a PFA reaction tube. Although the source produces lower HONO levels than demonstrated in the former study by Febo et al., I have several concerns with the manuscript.

Major Concerns:

1) Calibration source:

[Figure]

While I realize that the source can produce more or less constant HONO levels under certain conditions (see below) this is not a calibration source. If you want to use this source e.g. to calibrate a new HONO instrument, you need another instrument to characterize the source before under fixed experimental conditions. Even the HCl-concentration cannot be used to calculate the expected HONO level because of the parallel "H2O reaction", see below. Thus a HONO source is described here but not a calibration source! A calibration source could be e.g. the HONO source by Taira and Kanda (1990) using known Henry's law coefficients of HONO in the acidic bubble solutions and measured gas and liquid flows (NaNO2/H2SO4).

2) Stability of the source:

Already in the title the stability of the HONO source is high-lighted. However, when looking into detail to the study, I cannot see this stability, which is much lower than in the original study by Febo et al., but also lower than in several other studies.

Already the HCl production of the permeation source is not very stable and the variations are not well described, cf. e.g. Fig. 4a and Fig. S7. Also when you compare the HCl emission of the different PDs listed in Tab. 1, 2, 3 variations are observed which are much larger than the specified singles errors. E.g. In Tab. 1 the source PD-6a emits at 40 °C 0.21+-0.01 ppb HCl, whereas in Tab. 2 and Tab 3 0.08+-0.002 ppb and 0.33+-0.007 ppb are listed? What are the reasons for this variability?

Also the stability of the HONO source is very low. E.g., it takes more than 24 hours until the source produces stable HONO, see Run 1 in Fig. 3. Or why is the HONO source so unstable over 90 hours in Fig. 4b? These spikes are not really well explained and do not completely disappear even after long running time (see Fig. S10, Run 2). And even when the source is running for long time a variability of the HONO emission of 24 % is specified (see Figure 5b and line 525). This means that even when the HONO source is calibrated with very high accuracy by an independent method before (e.g. by precise NOy-measurement using chemiluminescence and a carbonate denuder)

[Figure]

than the uncertainty of a field calibration using the source is only +-25 %. This is not sufficient for many applications, when e.g. daytime levels of HONO are compared with calculated PSS levels under urban conditions, when the PSS is only a factor of two lower than measured values (in this case the uncertainty of the "unknown daytime source" would be +-50 %!).

The original source by Febo was much more stable and could also in theory be tuned to lower HONO concentrations by using more dilute HCl solutions (not tested in Febo et al., but works. . .). Also the simple source by Taira and Kanda is much more stable and shows precisions of a few percent. This source was further modified by using a temperature controlled stripping coil and also shows much higher stability (see https://quma-shop.de/images/LOPAP%2003%20HONO%20Source%20short_v2.pdf). I used this source during the FIONA intercomparison (http://euphore.es/fiona/fiona.html), which is a 19" instrument producing stable low HONO levels (ca. 1 % precision in the low ppb range) in a few tens of minutes which can be quickly and exactly tuned in a few minutes simply by changing the nitrite solution.

3) Purity of the source: In the present study a purity of >90 % is mentioned and impurities of NO of 5.5%/6 % are specified (lines 656 and 658). Typically NO impurities are formed by the bimolecular decomposition of HONO on surfaces producing equal amounts of NO and $NO_2$ (R3). Thus for me at least the NOx impurities are higher than 10 %, which is much higher than in the original paper by Febo et al. where NOx impurities of <0.5 % were determined at <20 ppm. Also the loss of HCl is not negligible. There is a clear small Cl- signal in Figure 2 with the source on (I expect this is of the order of few percent of the HONO level, please expand the Cl- signal). In contrast the simpler source by Taira and Kanda will not show HCl emission, since non-volatile $H_2SO_4$ can be used.

4) Formation mechanisms: In the present study two formation mechanisms are identified, the "HCl-" and the "H2O-mechanism". While the acid displacement mechanism of HONO by the strong acid HCl is clear (see Febo et al.) the H2O-mechanism should

be better characterised (see line 406: ". . .beyond the scope of this work.") since this is the main reason for the low stability of the HONO source for the initial phase when the source is started (e.g. first 24 hours for Run 1 in Figure 3, or »40 hours in Figure 6, the final steady state level will be 5 ppb in Fig. 6, see HCl emission of PD-6c in Table 1). The authors propose that NaOH is formed when $H_2O$ reacts with the sodium nitrite (see line 405). However, I expect that they still have $CO_2$ from their zero air generator which decrease the pH in the adsorbed water surface layers on the $NaNO_2$ to pH = 5.5 at which the fraction of HONO to nitrite is ca. 0.5 % (see pKa of HONO) making small HONO emissions possible in the absence of HCl. With extended operation of the source the increasing amount of $NaHCO_3$ and $Na_2CO_3$ formed on the surface will decrease the pH leading to lower HONO formation. This mechanism could also explain the much higher HONO emissions of the annular denuder compared to the PFA-tubes when using the same HCl permeation source (see the 6.2 ppb in Table 2 compared to the 24.5 ppb, see line 626, or the 400 -> 25 ppb in Fig. 6, when using PD-6c). Only when this non-stable side reaction can be minimized there is a chance for a really stable and predictable (by [HCl]g) HONO source!

Specific Concerns:

The following concerns are listed in the order how they appear in the manuscript.

Line 22: while 90% purity is possible, I do not see the 99% (see above)!

Lines 42-43: HONO levels of up to 18 ppb have been determined in the Po-valley (Milan) in Los Angles and in Santiago de Chile.

Line 46: Start with 2010a when using references by Veres et al. for the first time (and 2010b in line 113).

Line 66: The study by Villena et al., 2011 is not on HONO?

Line 80: A modified HONO source of the one by Taira and Kanda is missing (Kleffmann et al., J. Phys. Chem. A, 2004, 108, 5793-5799, see also link above).

[Figure]

Line 85: The advantage of the source by Taira and Kanda is that non-volatile dilute H2SO4 can be used bearing no risk of HCl emissions for incomplete acid displacement.

Line 104: The reason why the nitrite was stirred in Febo et al. is a) to minimize concentration peaks of HONO in channels in the nitrite powder to reduce the quadratic reaction kinetics of R3 and b) to increase the assessable surface area for complete acid displacement.

Line 109-110: The stabilization of the original HONO source by Febo et al. takes much shorter than the present source (see Febo et al., 1995). When I used that source several years ago it took only ca. 1 hour to get stable HONO levels, e.g. when the temperature of the HCl bath was changed. And other sources are even faster (see major concerns).

Section 2.1 and 3.1: The coating procedure of the PFA tubes is not clear for me. In section 2.1 it seems that all the nitrite added by the 3 ml coating solution stay inside the tube until the solvents are completely evaporated. However in Section 3.1 only a small fraction of the nitrite is recovered. Where is the rest? Or is the solution decanted similar to the procedure used for the annular denuder (see line 158)?

Figure 1 and S3: Are each two holes (1/2") in the aluminium block (see S3) used for one HONO source (1x HCl and 1x NaNO2, see Figure 1), i.e. can two parallel HONO source lines by used here?

Section 2.3 and S2: The set-up of the HCl permeation source is not clear for me. Is the 1/4" tube filled with liquid HCl inserted into a 1/2" PFA tube which is temperature controlled in the Al-block and flushed by the zero air? Is the smaller tube fixed in the middle of the larger PFA tube (how?) or is it simply lying on the ground of the larger tube? Please show a more detailed figure of the HCl permeation source.

Line 209 and Fig.1: Why is the low HONO source flow rate of only 100 sccm used and later diluted to 1.1 slm? If the source should be tuned to high HONO levels (ppms, see

line 664), the HONO concentration in these 100 sccm are 11 times higher leading to stronger decomposition of HONO by R3. Why not using 1.1 slm for the reaction?

Line 214: Typically also 1 % glycerol is added to a coating solution of a denuder (see coating of the $NaNO_2$ reaction devices, see line139)?

Section 3.2: caused by the non stable reaction by "$H_2O$" (or $CO_2$?) this side reaction should be minimized, see major concern. I cannot follow the statement in lines 335-337 – HONO production by this reaction is not defined and not constant, see Figure 6!

Lines 350-353. For me the source is already unstable at 40 °C, see peaks at 2 PM in Figure S7.

Table 1 and section S2: The production of the HCl tubes seems to be really "tricky" since similar tubes produce variable HCl emission by one order of magnitude! In addition the given errors of 0.01 ppb for the first three HCl concentrations are only a short term precision but not for longer time, compare data in Tables 2+3.

Line 380-384: Why are the HCl emissions decreasing by one order of magnitude for a time period of 2 years? Is this in accordance with the expected decrease of the liquid HCl concentration (e.g. 6 M) from the loss of gaseous HCl? In addition, a 30 % decrease is not similar to an order of magnitude (2 -> 0.21. . .).

Table 2: ADEs of only >55% and >84 % can be determined based on the given errors of HONO (e.g. 0.14/0.31. . .) and not >99%.

Figure 2: The red shaded area is almost invisible during zero. In addition I do not see any blue shaded area?

Line 458-459: That is not true for Run1 in Figure 3 (>24 h. . .)

Figure 3, 5, S9, S10, S12: Why do the authors not use a more precise HONO instrument or higher HONO levels? The precision of 0.4 ppb of the NOy monitor is too low to determine any dependencies at HONO levels of ca. 2 ppb. Typically the concentrations

should be more than 10 times higher than the precision of the instrument used. . .

Line 477 and Figure 4: I do not understand the low stability of the HCl/HONO pro-
duction (variability by factors, see Figure 4b. . .)? From where are these peaks coming
even after 15 days of use, see Figure S10?

Line 501: should be "after" and not "before" or "Figure 4b-c"? In Fig 5 the output is
more or less stable (+-25%...).

Line 505-506: I cannot follow that statement, see major concerns.

Line 563-565: Both data shown in Figure S10 are not stable (+- factor of 2) and I cannot
follow the RSD of only 16.5%? Cf. the red data at 5, 10 and 37 h.

Line 592: The relative stability of the HONO source is lower at 40 °C compared to
50°C see Table 3 (1+-0.5 = +-50% and 2+-0.5 = +-25 %). So I do not understand that
statement?

Line 606-607 and Table 4: I do not understand the lower output of the quartz tubing?
Normally quartz is hydrophilic and can be much better wetted by aqueous solutions
than the very hydrophobic PFA, leading to higher expected nitrite levels for the quartz?

Lines 624-630: The reason for the unstable and much higher HONO emissions (»[HCl])
is the higher surface area and the unwanted side reaction by "H2O" (or CO2?), see
above. . .

Line 643-644: The source by Febo et al. was much more pure (>99.5%) than the
present one (see 6 % NO = 12 % NOx, see R3 => purity 88%...).

Line 662-664: While the former study showed impurities in the 10 % range at 4 ppm,
this source shows similar impurities already below 1 ppb (see line 657), which is very
untypical with respect to the quadratic reaction kinetics of reaction R3. And again in
Febo et al. the purity was higher than 90% even in the ppm range.

Line 675: Should be <25 % see line 560?

Line 858: Processes

Line 887 and 902: …Res. Atmos.

Line 988: 10155-10171

Line 993: 9093-9106

Line 999: delete the second Vecera, Z.

Line 1010: D23302

Line 1061: L15820

Line 1074: 1681

Supplement Section S1, second paragraph, line 5: should be 47 and 51 sccm, see Figure 1?

Section S1, last paragraph before Figure S1: The temperature should be electronically limited to 150 °C (see harmful degradation products for PFA in the case of a malfunction…).

Figure S3: Why are larger hole diameters (0.6") used in the Al block for the 0.5" PFA permeation source tubes? There would be better heat transfer, if the holes would be only slightly larger than the PFA tubes (e.g. 0.51") getting them into direct contact with the aluminium.

Figure S7. From where do the peaks during the temperature ramps (40-50°C and 50-60°C) result? There should be constant increases of HCl similar to the ramp between 30-40 °C?
* * *

---

## Referee Comment (RC2) · Anonymous Referee #2 · 17 Jul 2020

Lao et al. present the design of a HONO calibration source suitable for field use. The manuscript gives a detailed description of the design of the source and extensive information on its performance. The paper solves a challenge that the community currently faces, the lack of reliable HONO calibrations of in-situ instrument. It is thus timely and highly relevant.

Overall the paper is very well written and informative. It provides all the details needed for the community to reproduce the study and, most importantly, build their own HONO calibration source. I did not find any real issues with the manuscript, and my more detailed comments below are just requests for some minor clarifications. Overall, the manuscript is well suited for publication in AMT, and I recommend for publication after

a few minor clarifications.

- As water bath operates at 40C producing a RH of 50%, I am wondering if there is an issue with condensation once the air flow cools down while exiting the oven and/or instrument. This may be especially relevant if the system is used in cold climates. Could you please comment on this in the manuscript?

- Could you explain in the manuscript how you determined relative humidity? Was it measured or calculated from the saturation vapor pressure and the gas flows? In the latter case, did you check that the calculated RH is in fact correct?

- You mention that the system has been found to be stable for approximately one month. Could you clarify if this means that it yielded a constant HONO concentration for this time period, or was there a drift in the concentration due to aging of the $NaNO_2$ reaction device?

  Could you comment on whether the length of the tubing between the source and the HONO instrument (or NOx instrument) has an impact?

---

## Referee Comment (RC3) · Anonymous Referee #3 · 29 Jul 2020

Lao et al. presented the design and detailed test results of a system capable of producing a wide range of concentrations of gas-phase HONO. The system is a modification to that by Febo et al (1995), based on displacement of nitrous acid from solid sodium nitrite by gaseous hydrochloric acid. The modifications include the uses of a custom-built HCl permeation source and a NaNO2-coated tube (or denuder) for the HONO displacement to take place. These modifications indeed make the system more compact in size and easier to transport. The tests are comprehensive, the results are well presented, and manuscript is well prepared. The manuscript can be helpful for laboratories in constructing a portable HONO generation system for calibrating field HONO instruments and for atmospheric HONO chemistry research in the laboratory. However, I find following major issues with the manuscript:

[Figure]

1. Compared to the original design by Febo (1995), the system described in this manuscript offers more compact in size and is capable of generating lower HONO source at concentrations (sub ppb vs 5 ppb). However, there are no improvements in stability and reproducibility (∼24% vs <0.4%), and purity (>90% vs 99.5%) of the HONO source generation. The NOx analyzer used to quantify HONO output has a lower detection limit of ∼0.4 ppb. The poor signal stability of the NOx analyzer near its detection limit may in part be responsible for the not-so-great performance of the system.

2. The HONO gas stream generated by this system must be calibrated by a primary instrument before it could be used as a HONO calibration source. Therefore, the system itself is not a tunable calibration source for HONO as the authors claimed in the title. A better HONO measurement will be needed to calibration the HONO source at low concentrations; the NOx analyzer described in the manuscript is not adequate for this purpose.

3. A commercial HONO source is available, based on the reaction of NaNO2 with diluted H2SO4 solution in a stripping coil (Taira and Kanda, 1990; Kleffmannet al., 2004) (https://quma-shop.de/images/LOPAP%2003%20HONO%20Source%20short_v2.pdf). A unit was used during the FIONA intercomparison in 2010 in Valencia, Spain, to generate a HONO source being distributed to and shared by the collaborating groups. Based on my own experience from this intercomparison, the system appeared to perform significantly better than the system described in the manuscript. Specifically, it offered high precision and stability (∼ 1% at low-ppb concentrations) and is truly tunable (stabilized within minutes after switching to a different concentration). Of course, the commercial unit could be much pricier than the home-build system described in this manuscript.

References

Febo, A., Perrino, C., Gherardi, M., and Sparapani, R.: Evaluation of a high-purity and high stability continuous generation system for nitrous acid, Environ. Sci. Technol., 29(9), 2390–2395, doi:10.1021/es00009a035, 1995.

Kleffmannet, J., Benter, T., and Wiesen, P.: Heterogeneous reaction of nitric acid with nitric oxide on glass surfaces under simulated atmospheric conditions. J. Phys. Chem. A, 108, 5793-5799, doi.org/10.1021/jp040184u, 2004.

Taira, M., and Kanda, Y.: Continuous generation system for low-concentration gaseous nitrous acid, Anal. Chem., 62, 630-633, 1990.
* * *

---

## Author Comment (AC1) · 16 Aug 2020

Due to the numerous comments from the three Reviewers, along with overlapping concerns, we have compiled a duplicate response in the attached PDF document that is colour-coded to denote responses (yellow) and manuscript modifications (green) to facilitate rapid review. We provide the responses here without the visual guidance, denoting Reviewer Comments with (RC) and our Author Responses with (AR). We provide the detailed response to all three Reviewers here as there are overlapping concerns.

General Response:

We thank the Referees for their time to provide comments and feedback on the

manuscript. We have edited the manuscript to address and incorporate their suggestions. In many cases, we realize that the clarity of our writing could be improved to address comments where the Reviewers had to make assumptions about technical details of our technique. We think that the manuscript contents are now clear, reflect the necessary corrections, and that the work is improved to a state we think meets the expectations for publication in Atmospheric Measurement Techniques.

Anonymous Referee #1

(RC) In the manuscript by Lao et al. the development and characterisation of a portable calibration source of HONO is described. The generation of HONO is based on a former source developed by Febo et al. (1995) and which is based on a reaction of gaseous humidified HCl with solid sodium nitrite. In contrast to the former study HCl is produced by a modified custom-built permeation source and the sodium nitrite is simply deposited on a PFA reaction tube. Although the source produces lower HONO levels than demonstrated in the former study by Febo et al., I have several concerns with the manuscript.

(AR) We thank the Reviewer for their thorough and considered comments on our manuscript. Many of the major concerns seem to have arisen from a lack of clarity which we have attempted to rectify throughout the manuscript. In a number of instances, the Reviewer has overlooked significant sections of our submitted work that address the concerns expressed in their comments and we have made significant effort to draw their attention to our existing discussion below, with changes made to the manuscript where improved clarity was beneficial to resolving the expressed concerns.

(RC) Major Concerns: (RC) 1) Calibration source (RC) While I realize that the source can produce more or less constant HONO levels under certain conditions (see below) this is not a calibration source. If you want to use this source e.g. to calibrate a new HONO instrument, you need another instrument to characterize the source before under fixed experimental conditions. Even the HCl concentration cannot be used to

calculate the expected HONO level because of the parallel "H2O reaction", see below. Thus a HONO source is described here but not a calibration source! A calibration source could be e.g. the HONO source by Taira and Kanda (1990) using known Henry's law coefficients of HONO in the acidic bubble solutions and measured gas and liquid flows (NaNO2/H2SO4).

(AR) The point of the Reviewer regarding appropriate use of nomenclature is understandable and we are disappointed that they find our terminology misleading. However, first principles considerations suggest that the use of the word calibration in their example is also incorrect. Calibration refers to determining the response of an instrument when detecting a known quantity of an analyte. The method by which a given quantity of analyte is determined therefore must exist for all calibration approaches. The NaNO2 concentration and H2OS4 concentrations, in the example provided by the Reviewer, would have to be validated against a primary standard or be prepared as primary standards themselves, as evidenced in the report provided by the Reviewer below where they reported use of ion chromatography with UV-Vis detection for just such a purpose (Kleffmann et al., 2004). Further, uncertainty in the Henry's Law coefficient for HONO, and the flows of the NaNO2 and H2SO4 solutions would need to be propagated into determining the HONO quantity generated by the calibration approach suggested by the Reviewer. All calibrations are subject to uncertainty in the analyte quantity and require validation against some benchmark. Thus, validation of analyte concentrations completed in the laboratory by a NOx analyzer prior to use in the field for our HONO source, or the aqueous solutions the Reviewer suggests, are synonymous calibration approaches. Notably, without the need to quantify the emission rate of HCl from a permeation device (or to use a cylinder or thermostated bath of HCl in the field), generating HONO from water vapour and a coated NaNO2 reaction device in our work reduces the complexity of this task, as the NOx impurities can be ignored in the absence of requiring mass balance between HCl delivered versus HONO produced.

(RC) 2) Stability of the source: Already in the title the stability of the HONO source is

high-lighted. However, when looking into detail to the study, I cannot see this stability, which is much lower than in the original study by Febo et al., but also lower than in several other studies.

(AR) The Reviewer is concerned with the claim that the presented HONO source output is stable, contending that it is not so. We feel that several sections of our manuscript (See Sections 3.5 and 3.6)) have been overlooked by the Reviewer to suggest our HONO source is not reliable. The section discussions are comprehensive in the manuscript to guide the community on how to achieve stable output, along with some advice on issues that may be easily overlooked (e.g. transient 'burbs' of HCl from permeation devices) when attempting high accuracy calibrations. We rebut these points in detail here and indicate changes where we believe clarity in the text can be improved.

We reiterate the stability statistics quantified by four separate research grade instruments all validated to measure HONO accurately to demonstrate the stability of our methodology, from Lines 471-481:

'Three trials using newly coated $NaNO_2$ reaction devices and PD-6a, once stabilized, generated an average HONO output of 2.28±0.58 ppbv, which corresponds to an RSD of 24 % between runs and an RSE of 3% (n = 2367; Figure 3). The noise observed in the stabilized HONO output in Figure 3 can be primarily attributed to the noise associated with the $NO_x$ analyzer detector (18 of the 24 %; DL = 0.4 ppbv; 1-minute average). This conclusion is supported by the lower noise in $\sim$2.5 ppbv HONO mixing ratios observed by the CIMS (Fig. 2, RSD of 8.1%), ACES (RSD 8.2 %), and NOyO3 (RSD 1.9 %). In these added observations with higher sensitivity instrumentation, the stability was equal to instrumental precision. This represents a major improvement over our previously reported calibration sources with potential for 30 % variability at a minimum (VandenBoer et al., 2013; Zhou et al., 2018).'

(RC) Already the HCl production of the permeation source is not very stable and the

variations are not well described, cf. e.g. Fig. 4a and Fig. S7. Also when you compare the HCl emission of the different PDs listed in Tab. 1, 2, 3 variations are observed which are much larger than the specified singles errors. E.g. In Tab. 1 the source PD-6a emits at 40 _C 0.21+-0.01 ppb HCl, whereas in Tab. 2 and Tab 3 0.08+-0.002 ppb and 0.33+-0.007 ppb are listed? What are the reasons for this variability?

(AR) We thank the Reviewer for bringing up these important points. The figures mentioned here are presenting HCl emissions where we are rapidly altering conditions, such as temperature (Fig. S7), to demonstrate the time required to regain stable output (e.g. see error bars on Fig. S6). In Figure 4a, we are showing transient issues in the use of permeation devices, which are not captured from standard certification practices for their emission rates and may affect calibration accuracy when using the permeation device approach to produce higher mixing ratios of HONO on the order of several ppbv. We did not present any plots of the stable output for our devices, as a straight line of 0.21+/-0.01 ppbv is not highlighting any issue of significance. For these reasons, it is understandable that the Reviewer feels that there is a conflict in the presentation of the use of the HCl permeation devices. It was indeed our intention to highlight some of the unexpected results we encountered using them, which we wish to present to the community transparently.

Overall, it is important to note the issues when using custom permeation devices in a HONO source. Their emission rates do change over time, such that they are not solely dependent on the concentration of solution used, the surface area and thickness of permeable polymer, and the temperature (O'Keeffe and Ortman, 1966; Susaya et al., 2012). Therefore, we do not expect every custom permeation device we make to always have the same emission. This is broadly true for commercial permeation devices and includes larger emission uncertainty (see excerpt from manuscript below). Reliance on certified commercial devices comes with both a large cost (∼hundreds of dollars) and time investment (∼6 weeks) so that emission rates can be determined by mass difference, with the emission rate values having much higher uncertainty due to

the indirect nature of quantifying the active compound.

We addressed the variability in our custom HCl permeation device emission rates, including relevant context to commercial options, in detail in Section 3.3, including Table 1 and Lines 379-391:

'Two newly made 6 M HCl PDs (-6b and -6c) were found to emit different, yet highly stable (e.g. ±0.01 ppbv), mixing ratios at identical oven temperatures (Figure S6). This demonstrates potential variability with each new device due to inconsistent results during custom fabrication compared to commercial PDs. The most likely source of such differences in output is variability in our sealing of the PTFE plugs resulting in increased emission rates. In any case, the PDs remain stable with less than 10 % relative standard deviation. In comparison, commercial device emission rates are often certified within ±30 %. The emission rates of commercial PDs are certified through measurement by gravimetric weight loss over time (ng min-1). A commercial $\frac{1}{4}$" (64 mm) Teflon HCl PDs of 6.55 M certified to emit 1905 ± 520 ppbv in 100 sccm flow at 40 ËŽC (RSD = 27.3 %; VICI Metronics, Inc.; Poulsbo, WA), has this output variance due to the co-emission of water and propagated measurement uncertainties. A lower variance in the emitted HCl was observed from our custom-made PDs when we quantified HCl directly by either CRDS or IC-CD. Custom-built PDs were therefore chosen over commercial PDs due to their demonstrated stability and low cost.'

We have also highlighted the reasons for the variability in the output of a single permeation device in the manuscript, which the Reviewer seems to have missed. We discuss the observed decline in the output of PD-6a in detail, with reference to existing literature, at Lines 392-400:

'It was found that HCl outputs of the custom-PDs slowly diminished over time, which emphasizes the need for regular calibrations. For example, the HCl output from two-year-old PD-6a emitted 0.21±0.01 ppbv in 1.1 SLPM in comparison to 2.0±0.01 ppbv when it was newly made, which decreased the resulting HONO generation in the reaction device. Similar results have been observed in calibrations with PDs of aqueous NH3 and HNO3 solutions decreasing by ∼30 % during two years of storage, as well as for carbonyl sulfide (Fried et al., 1998; Neuman et al., 2003). Despite the decreasing HCl output over a year or more of use, HCl PDs act as a stable acid source on the order of weeks, producing consistent HCl output to subsequently generate stable HONO, even when removed from the permeation oven or stored for up to two months.'

Therefore, we feel that further manuscript additions to explain that the same temporal decline in emissions accounts for the continued decrease in emission of HCl from PD-6a presented in Table 2 is not necessary. The emission rate quantified in Table 3 clearly pertains to the section of the manuscript presented above. We feel that this is clear in the main body of the manuscript and have not made modifications.

However, we have added additional text to the conclusions to highlight to the reader best practice when using custom-made HCl PDs at Lines 735-737:

'Custom-made HCl PDs are prone to variability in emission rates, both between similarly made PD and over time, and therefore require regular characterization, but can provide a stable output over the order of weeks'.

(RC) Also the stability of the HONO source is very low. E.g., it takes more than 24 hours until the source produces stable HONO, see Run 1 in Fig. 3.

(AR) This comment is confusing as the Reviewer seems to have issue with the rate to which a stable output is reached, and not the stability characteristics of the HONO output itself, which we have already addressed above. The Reviewer has missed a clear explanation of the observation in our discussion at Lines 470-473:

'The time required to achieve stable HONO signals was tested by inserting HCl PD-6a and new NaNO2 PFA reaction device into the calibration system, followed by flow start up. Stable HONO signals were observed within 7 h of powering on the HONO calibration system. This is 5 h longer than required to reach stable mixing ratios for a

previously stabilized NaNO2 device.'

The device reached stable output within 7 hours, not 24 as the Reviewer suggests, and this is clearly stated in Section 3.5. We depict 24 hours of data in Fig. 3 to show the stability of the HONO produced and how an identical HONO output is rapidly reached with a number of freshly coated NaNO2 devices. In the best example, this is reached in 2 hours for Run 3. We also state here that when an NaNO2 device is stabilized once, that it can produce stable HONO output again within 2 hours of starting the system henceforth. There is an entire section of our manuscript devoted to this (Section 3.6) and it is clearly depicted in the 8 runs presented in Figure 5.

To improve clarity, we have changed our reference to the 'runs' presented in Figure 3 to now state 'device' and added a vertical dashed line to the figure to highlight where the output of all three coated NaNO2 devices are at the same stable HONO output (determined statistically), as described in the discussion, to improve the connection of our discussion to the figure. We have attached the revised figure and present the revised caption here:

'Figure 3. Mixing ratios of HONO observed using HCl PD-6a and three different, but freshly coated, NaNO2 PFA reaction devices. Time zero indicates the start of HONO production in the calibration unit where no prior flow through the calibration unit existed, but all temperatures were stable at 40 °C. The vertical dashed line denotes the time where the output of the three devices are no longer statistically different from each other. Reported measurements are one-minute average data with a 30 s Kalman filter on the NOx analyzer.'

(RC) Or why is the HONO source so unstable over 90 hours in Fig. 4b? These spikes are not really well explained and do not completely disappear even after long running time (see Fig. S10, Run 2).

(AR) Here again, we cannot provide more information than the extensive section of the manuscript that has already been written to discuss this from Lines 492-498 and

508-517 (our emphasis added in bolded italics below):

'However, when the HCl output from PDs is unstable, this can interfere with the stability of the HONO generated because it is dependent on acid displacement. A common characteristic of our custom-PDs monitored by real-time CRDS measurements are short-duration increases in output over min, up to 1 h, due to reduced emission of $H_2O$ and increased emission of HCl, resulting in transient pulses from the device (Figure 4a). The anticorrelation between HCl and $H_2O$ is expected for a constant mass emission to result from the contained aqueous solution. A corresponding rapid increase in HONO production results from such occurrences (Figure 4b).

Commercial PD manufacturers evaluate average mass emission rates by gravimetric weight loss over several weeks at 40 °C for certification, which could include such short-term events. The HONO output from a newly made custom-HCl PD (PD-6c) over four consecutive observation periods upon insertion of a new $NaNO_2$ reaction device (Runs 1 − 4) at constant temperature (40 °C) show that the new custom-PD requires about 1 week of operation before its output is stable (Figure 4b-c). Therefore, careful preparation of PDs and $NaNO_2$ reaction devices in advance of extensive use will yield a HONO calibration source with the fastest stabilization times possible for continuous operation over a period of months. Note again, that the HONO measurements for Figures 5b-c were performed several months before the HCl emission rate for PD-6c presented in Table 1 were obtained, resulting in high HONO mixing ratios produced in these experiments.'

The comment regarding Figure S10 is explained, from the newly prepared nature of the custom permeation device, in the main body of the manuscript from Lines 577-584, where it is referenced:

'We next tested the reproducibility for newly made HCl PDs. Two experiments used PD-6b, containing 6 M HCl (Figure S10). After a period of stabilization, the two experiments generated similar HONO mixing ratios (2.58±0.43 ppbv after 25 hrs, RSD =

16.5 %, RSE = 1.43 %, n = 792). The spikes in HONO output at 15 h and 21 h in the second experiment (green trace, Fig. S10) were likely due to pulses of HCl which we commonly observed with new PDs (e.g. Fig. 4a-b). This emphasizes our recommendation that new custom-HCl PDs should be used for an extended period prior to use for acid displacement to ensure the emission rate has stabilized.'

(RC) And even when the source is running for long time a variability of the HONO emission of 24 % is specified (see Figure 5b and line 525). This means that even when the HONO source is calibrated with very high accuracy by an independent method before (e.g. by precise NOy-measurement using chemiluminescence and a carbonate denuder) than the uncertainty of a field calibration using the source is only +-25 %. This is not sufficient for many applications, when e.g. daytime levels of HONO are compared with calculated PSS levels under urban conditions, when the PSS is only a factor of two lower than measured values (in this case the uncertainty of the "unknown daytime source" would be +-50 %!).

(AR) We have already addressed this comment above. As we demonstrate in the manuscript it is quite possible to achieve less than 10 % variance between restarts of the instrument, yet we report 24 % as a lower limit of the system performance based on our experience and the use of a low precision instrument (NOx analyzer, DL = 0.4 ppbv). We note here, and throughout the manuscript, that no field-deployed gaseous HONO calibration sources from the recent literature have achieved such performance; they all rely on laboratory calibrations that have similar uncertainties on the order of 25 % when validated against an external orthogonal method (McGrath et al., 2019; Peng et al., 2020; VandenBoer et al., 2013; Young et al., 2012; Zhou et al., 2018). The use of aqueous nitrite standards has long been the method of wet chemical instrumentation, resulting in high precision, but these do not represent overall method accuracy. The total instrument accuracy would necessarily include an inlet addition of HONO, which is not standard practice (Afif et al., 2016; Crilley et al., 2019; Pinto et al., 2014; Ren et al., 2011; VandenBoer et al., 2014). Even the gold standard of differential

optical absorption spectroscopy relies on a calibrated cross-section determined using long-duration observations of HONO from the Febo et al. (1995) calibration source at high mixing ratios, to minimize error (Stutz et al., 2000). To our knowledge, there are no reports of HONO calibration sources in the literature that achieve better than 25 % accuracy performance at environmentally relevant levels. We note that most reports HONO measurements report variance in the measurements without including the accuracy of the measurements themselves. This is certainly a concern for the HONO community in assessing the HONO PSS.

Regarding stability of HONO production we note here, prior to the additional comments from the Reviewer below, that Febo et al. (1995) do not specify the averaging time for their reported stability numbers. Based on the instrumentation used, the mixing ratios (e.g. Fig 3 at $\sim$ 7 ppmv!), and the presented data, we have concluded that the reported variance is most likely based on 1 minute measurements averaged over at least 12 hours, for which our % RSE values of 3 % on similar timescales compare very well given the >3 orders of magnitude lower mixing ratios we measured. Thus, the system presented in this work not only provides stable HONO mixing ratios but does so at considerably lower levels of environmental relevance.

(RC) The original source by Febo was much more stable and could also in theory be tuned to lower HONO concentrations by using more dilute HCl solutions (not tested in Febo et al., but works). Also the simple source by Taira and Kanda is much more stable and shows precisions of a few percent. This source was further modified by using a temperature controlled stripping coil and also shows much higher stability (see https://qumashop. de/images/LOPAP%2003%20HONO%20Source%20short_v2.pdf). I used this source during the FIONA intercomparison (http://euphore.es/fiona/fiona.html), which is a 19" instrument producing stable low HONO levels (ca. 1 % precision in the low ppb range) in a few tens of minutes which can be quickly and exactly tuned in a few minutes simply by changing the nitrite solution.

(AR) We note that as we are using permeation devices, which always require calibration or certification, that the resulting system is much safer and simpler to use over mixing liquid solutions of acid. We choose to use permeation devices as they are portable and easy to use in the field and have been successfully used for field calibrations of numerous other compounds (Veres et al., 2010; Washenfelder et al., 2003). Neither the method of Taira and Kanda (1990), nor the modified version published by the Reviewer below (Kleffmann et al., 2004), (1995) has achieved wide use in the atmospheric chemistry community (compared to that of Febo et al. (1995)), presumably due to the need for a peristaltic pump and custom glassware or the cost of the commercial unit from QUMA mentioned. Further, the mixing ratios of HONO generated by the Taira and Kanda (1990), Kleffman et al. (2004), and the Febo et al. (1995) approaches are often much higher than found in the ambient environment. These require several expensive and power-demanding gas pumps, mass flow controllers, and dilution to reach single parts per billion mixing ratios and have not been demonstrated to reach sub-ppbv levels except by us (VandenBoer et al., 2013). From our personal experience with our own sources, the output was not stable within 25 % across multiple days and also required independent verification of the output by a gold catalyst NOy instrument (calibrated for NO and NO2 conversion independently with a certified cylinder) or ion chromatograph in order to accurately calibrate instrumentation to better than 10 % for use in either the lab or the field (McGrath et al., 2019; VandenBoer et al., 2013, 2015; Zhou et al., 2018). We emphasize again our goal of developing robust instrumentation, which means that all instances of moving parts should be avoided, chemical safety hazards minimized (i.e. no bulk solutions), and the resulting system should be easily assembled and operated by users with a wide range of expertise.

Since the QUMA calibration source performance is a personal statement on observations that have not been subject to peer-review, we can only congratulate the Reviewer on their modified source performance and encourage them to publish this work and the results of the FIONA intercomparison. We have investigated the published work referenced by the Reviewer below that is related to this major concern, which reports a

modified source that was capable of generating 30-770 ppbv of HONO. However, the rapidity of the transitions, precision, and accuracy of the output are not provided in the manuscript (Kleffmann et al., 2004). The output of the reported source in the work of the Reviewer was quantified by ion chromatography, a technique used in nearly all our prior work with HONO calibration sources (e.g. VandenBoer et al. (2015)), from which the accuracy is regularly on the order of 15 % when robustly assessed. Given that our source is stable to the precision of all instruments that measure its output at ~2 ppbv with a relative standard error of 3 % within 2 hours (120 minutes, comparable to 'tens of minutes') of being turned on, as stated above, perhaps analysis of our source output with higher precision instrumentation will be the subject of a future intercomparison, as we suggest in the Conclusions. We note here that adjusting the calibration mixing ratios with mass flow controllers in this system respond on the order of seconds, as has been well-established in our prior work (Roberts et al., 2010; VandenBoer et al., 2013). The versatility in changing output by orders of magnitude that the Reviewer is concerned with requiring a few hours is not a major trade-off. Our source inherently produces HONO at much lower mixing ratios than any others to date, such that the need for such adjustments is an infrequent demand. In response to further comments from the Reviewer below, we have made an addition to the Conclusions of the manuscript to note that the four channels in the Al-block of the permeation oven can be used to operate parallel HONO source channels and addressing the 'tunability' concern further.

We have added this excellent point to our discussion at Lines 723-725:

'The HONO calibration source was designed to facilitate multiple calibrant concentrations, as the four holes in the aluminium heating block (Fig. S3) allows for the operation of parallel HONO sources if desired.'

We would like to emphasize a formal point regarding peer review: existing published literature should be used to support concerns in our work submitted for review in Atmospheric Measurement Techniques over the Reviewer's opinion or unpublished results on their preferred calibration source, where a conflict of interest with a commercial

product not widely used by the HONO community seems to also exist.

(RC) 3) Purity of the source: In the present study a purity of >90 % is mentioned and impurities of NO of 5.5%/6 % are specified (lines 656 and 658). Typically NO impurities are formed by the bimolecular decomposition of HONO on surfaces producing equal amounts of NO and NO2 (R3). Thus for me at least the NOx impurities are higher than 10 %, which is much higher than in the original paper by Febo et al. where NOx impurities of <0.5 % were determined at <20 ppm.

(AR) As the Reviewer rightly mentioned, Febo et al. (1995) found that <0.5% of the HONO signal (<20 ppmv) was a NOx impurity and was attributed to R3. In our system, R3 cannot account for the observations of NOx. In Section 3.7.3, where we specifically address the purity of the HONO source, the HONO output was 770 pptv. If we assume that the impurity from R3 is linear at these lower mixing ratios, i.e. that 0.5% of the HONO signal is NOx due to R3 (as an upper limit), we would expect a NOx impurity of 3.85 pptv due to R3. This is notably lower than what we measured. If we take the high sensitivity NO measurements (Rollins et al., 2020), as this was the instrument with the lowest detection limit, we observed 42 pptv of NO, or 5.5% of the total HONO, well above that predicted due to R3 from Febo et al. The bimolecular decomposition reaction (R3) requires sufficient levels of HONO to be able to react with itself. We note that the HONO mixing ratios in the current work are 2-3 orders of magnitude lower than previous work (e.g. Febo et al. (1995)). Due to the second-order nature of this reaction, it may be that the levels of HONO are so low that R3 is not significant due to the square dependence on the number density of HONO, making our concentration-independent scaling for the upper-limit estimate presented above a very conservative estimate. As we observed higher levels of NO than expected from the upper limit calculation, it points to additional chemical pathways affecting the purity of the HONO output sources capable of reaching the sub-ppbv range, as stated at Lines 685-690.

Thus, the best way to constrain the NOx impurity is through measurements. Using multiple instruments, no NO2 impurities were observed in any of our experiments, even

where NO was above the NOx analyzer detection limits (e.g. NaNO2-coated denuder tests). Despite these observations, we again provided a conservative perspective on our assessment of source purity by utilizing the measurements of the ACES platform which has higher precision and accuracy than our NOx analyzer. Thus, the potential NO2 and NO impurities from our source were found to be less than or equal to 10 % of the total HONO, with the limit set by the precision of our instruments at the low mixing ratios generated, see Lines 668-673:

'The I- TOF CIMS is not sensitive to NO or NO2, so further measurements were made with our Mo-catalyst NOx analyzer, a cavity-enhanced absorption spectrometer (Min et al., 2016), and a gold-catalyst NOy instrument (Fahey et al., 1985; Fontijn et al., 1970; Ridley and Grahek, 1990; Ridley and Howlett, 1974; Ryerson et al., 1999), which determined that NO2 impurities were at or below 10 % of the generated HONO based on the detection precisions of the latter two instruments.'

We acknowledge that the stated purity of the system described in the current work is lower than stated in Febo et al. (1995), but as we were aiming to build a field portable and robust HONO source at atmospherically relevant concentrations (i.e. sub-ppbv), we feel that the trade-off in potentially higher NOx impurities is more than compensated for by the ease of using this instrument in the field.

To clarify this point, we have added the following text to the conclusions at Lines 729-732:

'The purity of HONO source was determined to be >90%, and while lower than previous work (99.5%, Febo et al. (1995)) this may be a consequence of previously unseen side reactions of increasing importance at the low HONO mixing ratios generated. We consider this an acceptable trade-off for a robust field deployable HONO source unit.'

And we have changed the description of our purity spanning 90-99 % at Lines 21-22 in the abstract to read:

'The calibration source developed in this work can generate HONO across the atmospherically relevant range and has high purity (>90 %)'

(RC) Also the loss of HCl is not negligible. There is a clear small Cl- signal in Figure 2 with the source on (I expect this is of the order of few percent of the HONO level, please expand the Cl- signal). In contrast the simpler source by Taira and Kanda will not show HCl emission, since non-volatile H2SO4 can be used.

(AR) With regard to the loss of HCl, from Fig 2, the Cl (m/z 35) counts were 37 $\pm$ 12 compared to 1100 $\pm$ 135 ncps for the HONO signal when the gas exiting the source was sampled, and so the Cl signal represents 3$\pm$1 % as an average breakthrough using the one sigma uncertainties to arrive at a worst-case estimate. The m/z 35 counts during the HONO measurement were indistinguishable from the zero measurement, which gives a 3-sigma detection limit of 39 ncps at m/z 35. We concluded that no detectable, let alone quantifiable, HCl was exiting the calibration source.

This is further supported in the mass balance experiments discussed in our work in Section 3.4.1, where we observed no HCl exiting the source when using the CRDS which has a superior sensitivity (3 sigma detection limit of 5 pptv) compared to the CIMS. See Lines 437-438:

'No HCl was observed to exit the devices, indicating unit acid displacement efficiency and reaching mass balance.'

In order to avoid confusion from other readers, we have modified the discussion in Section 3.4.2 from Lines 454-457, which initially stated:

'The signal of Cl- at m/z 35 remained constant near zero throughout this period, confirming again that the HCl from the PD was entirely consumed by the NaNO2 reaction device throughout the measurement period, consistent with the experiments presented above where no HCl was measured by the CRDS.'

This section now states more clearly:

[Figure]

'The signal of Cl- at m/z 35 was below the detection limit throughout this period, con-firming again that the HCl from the PD was entirely consumed by the NaNO2 reaction device throughout the measurement period, consistent with the experiments presented above where no HCl was measured by the CRDS.'

We also note also that the approach/reaction mechanism described by Taira and Kanda (1990) is not applicable to the approach presented the current work, as we used perme-ation devices to deliver the gas-phase acid required for the acid displacement reaction and therefore need a volatile compound (i.e. one cannot use dilute sulphuric acid in a PD). Even so, if HCl were to pass through the reaction bed it does not bias the ability to quantify the HONO output or the stability so long as the kinetics are constant. In ad-dition, we have shown that the H20 mechanism is not possible to ignore at low HONO mixing ratio production and requires more attention to detail than the 'HCl in equals HONO out' strategy used in high-output HONO sources (Febo et al., 1995; Gingerysty and Osthoff, 2020; Taira and Kanda, 1990). We challenge the Reviewer's assertion that the source based on acid displacement of H2SO4 is simpler, as it requires a peristaltic pump, glass components to mix the liquid reagents and a solution of H2SO4, which all represent added complexity/hazard and reduce the portability and robustness of that calibration source for field use, particularly on mobile platforms.

(RC) 4) Formation mechanisms: In the present study two formation mechanisms are identified, the "HCl-" and the "H2O-mechanism". While the acid displacement mecha-nism of HONO by the strong acid HCl is clear (see Febo et al.) the H2O-mechanism should be better characterised (see line 406: "beyond the scope of this work.") since this is the main reason for the low stability of the HONO source for the initial phase when the source is started (e.g. first 24 hours for Run 1 in Figure 3, or Âż40 hours in Figure 6, the final steady state level will be 5 ppb in Fig. 6, see HCl emission of PD-6c in Table 1). The authors propose that NaOH is formed when H2O reacts with the sodium nitrite (see line 405). However, I expect that they still have CO2 from their zero air generator which decrease the pH in the adsorbed water surface layers on the

NaNO2 to pH = 5.5 at which the fraction of HONO to nitrite is ca. 0.5 % (see pKa of HONO) making small HONO emissions possible in the absence of HCl. With extended operation of the source the increasing amount of NaHCO3 and Na2CO3 formed on the surface will decrease the pH leading to lower HONO formation. This mechanism could also explain the much higher HONO emissions of the annular denuder compared to the PFA-tubes when using the same HCl permeation source (see the 6.2 ppb in Table 2 compared to the 24.5 ppb, see line 626, or the 400 -> 25 ppb in Fig. 6, when using PD-6c). Only when this non-stable side reaction can be minimized there is a chance for a really stable and predictable (by [HCl]g) HONO source!

(AR) We did not use a zero air generator for generation of HONO in the calibration source, we used either zero air or nitrogen cylinders (at >99.998% purity). The zero air generator was used to provide dilution flow prior to instrument sampling.

We have clarified the description in the text from Lines 176-180, which now reads:

'Carrier gas flow through the permeation oven was provided by a compressed cylinder of zero air or nitrogen (Praxair; Air Ultra Zero, 99.999%, AI 0.0UZ-K; High Purity Nitrogen, 99.998 %, NI 4.8, Toronto, ON) but an in-situ zero air generator could also be used (e.g. Aadco Instruments Model 747-10, Cleves, OH; used only for dilution flows here) providing 20 psi of pressure to control the flow entering a four-way $\frac{1}{4}$" (64 mm) Swagelok cross fitting.'

Furthermore, the water in the bubbler originates from a heated reservoir on our deionised water system, which is engineered with the explicit purpose of minimizing uptake of CO2. The deionised water was replenished regularly and would have been rapidly purged of CO2/CO3-/CO32- by the high purity gases used to generate HONO. Therefore, we expect that there should be very transient, but more often an absence of, CO2 entering the source. In either case, the amount would be insufficient for the pH to decrease with use. It is worth noting that the Reviewer is relying on the concept of bulk solution pH in this argument, which does not apply so readily to surface films

and we caution them from making this extrapolation. Consequently, we do not think that a decrease in pH in the reaction device due to $CO_2$ uptake is a viable alternative hypothesis to explain the mechanism we ascribe to $H_2O$, nor does $CO_2$ affect the stability of our source. We do not think the $H_2O$ mechanism is the cause of the instability of the HONO source when initially turned on, but rather due to the emissions from the PD stabilizing, which is typical for these devices.

We did think long and hard about the actual mechanism for the $H_2O$ mechanism but could not come up with anything plausible. Despite this, as the experimental evidence was strong, we wanted to share this finding with the community as this may negate the need for HCl (or indeed any acid) in a HONO source. Future work will explore the chemistry underlying the $H_2O$ mechanism.

(RC)Specific Concerns: The following concerns are listed in the order how they appear in the manuscript.

(RC) Line 22: while 90% purity is possible, I do not see the 99% (see above)!

(AR) Please see our earlier response, we have edited this to read (>90 %)

(RC) Lines 42-43: HONO levels of up to 18 ppb have been determined in the Po-valley (Milan) in Los Angles and in Santiago de Chile.

(AR) We thank the Reviewer for the added locations where high HONO mixing ratios have been reported and have located the references we suspect they are aware of, but without the provision of the actual citations by the Reviewer. We apologize if a specific measurement they had in mind is not in the list. Regardless, this sentence is to provide an example – hence our use of 'such as' here - on the observed HONO levels in polluted environments, not to perform an exhaustive review of such observations.

We have updated the text to include the references we have found at Lines 42-45:

'. . .to 18 parts per billion by volume (ppbv) in polluted megacities such as Milan, Los Angeles, and Beijing (Elshorbany et al., 2009; Febo et al., 1996; Harris et al., 1982;

Tong et al., 2016; Zhang et al., 2019).'

(RC) Line 46: Start with 2010a when using references by Veres et al. for the first time (and 2010b in line 113).

(AR) This is a known issue with the reference template for AMT used in the Mendeley software and is subject to correction during typesetting. We thank the Reviewer for their attention to detail.

(RC) Line 66: The study by Villena et al., 2011 is not on HONO?

(AR) Our apologies, we meant to reference the identification of HONO partitioning to fog and dew water to form nitrite, as observed by Rubio et al. (2009) where Villena was a co-author on that work, which would result in a particulate nitrite interference.

(RC) Line 80: A modified HONO source of the one by Taira and Kanda is missing (Kleffmann et al., J. Phys. Chem. A, 2004, 108, 5793-5799, see also link above).

(AR) We have not added the modified Taira and Kanda (1990) HONO source at this point in the manuscript as there is inadequate description of the modifications and performance of the system to make it suitable to include here. We have also not referenced the web links provided above as they are not peer reviewed materials.

(RC) Line 85: The advantage of the source by Taira and Kanda is that non-volatile dilute H2SO4 can be used bearing no risk of HCl emissions for incomplete acid displacement.

(AR) We have added a comment in the introduction to this effect when discussing this paper and the subsequent modification by Kleffmann et al. (2004) at Lines 96-101:

'An alternative approach that utilised dilute H2SO4 for the acid displacement reaction with aqueous NaNO2 was outlined by Taira and Kanda (1990). While this approach was shown to generate a stable and tunable HONO output at hundreds of ppbv, it has not been widely adapted, likely due to the need for complex custom glassware and liquid flow control in the calibration apparatus and significant dilution to reach single

digit ppbv mixing ratios (Kleffmann et al., 2004).'

(RC) Line 104: The reason why the nitrite was stirred in Febo et al. is a) to minimize concentration peaks of HONO in channels in the nitrite powder to reduce the quadratic reaction kinetics of R3 and b) to increase the assessable surface area for complete acid displacement.

(AR) We have clarified this, and included the point on decomposition kinetics from the Reviewer, at Lines 109-112:

'Further, to reduce the variability in HONO output over time, the powdered NaNO2 bed requires continuous mixing to maintain equilibrium between the adsorbed HONO and carrier gas flowing over the salt bed minimize the production of NOx by R3, as well as a Teflon filter to prevent loss of NaNO2 powder by entrainment in the gas flow.'

(RC) Line 109-110: The stabilization of the original HONO source by Febo et al. takes much shorter than the present source (see Febo et al., 1995). When I used that source several years ago it took only ca. 1 hour to get stable HONO levels, e.g. when the temperature of the HCl bath was changed. And other sources are even faster (see major concerns).

(AR) The literature we are referencing here are 'other systems using dispersed NaNO2' which are not stirred (McGrath et al., 2019; Roberts et al., 2010; VandenBoer et al., 2013; Zhou et al., 2018). We have already addressed the issue about the stabilization time of our system being 2 hours above and refer the Reviewer back to our responses on their major concerns above.

(RC) Section 2.1 and 3.1: The coating procedure of the PFA tubes is not clear for me. In section 2.1 it seems that all the nitrite added by the 3 ml coating solution stay inside the tube until the solvents are completely evaporated. However in Section 3.1 only a small fraction of the nitrite is recovered. Where is the rest? Or is the solution decanted similar to the procedure used for the annular denuder (see line 158)?

(AR) We have carefully stated where we believe the losses of the nitrite from the coating solution are in our procedure in both of these sections and refer the Reviewer to our statements in these sections. The solution is not decanted. The nitrite from the applied solution is partly lost during the coating process due to the highly hydrophobic nature of the PFA tubing which we have stated in Section 2.1 (Lines 154: 'to reduce solution loss while evaporating solvent') and Section 3.1 (Lines 287: '... and the loss of liquid solution during the drying procedure...'). Some of the nitrite may also be lost as NO, NO2 or HONO during the drying procedure, but an experiment was not conducted to explore this as we only required sufficient nitrite to be available on the surface to facilitate HONO production.

(RC) Figure 1 and S3: Are each two holes (1/2") in the aluminium block (see S3) used for one HONO source (1x HCl and 1x NaNO2, see Figure 1), i.e. can two parallel HONO source lines by used here?

(AR) Yes! As discussed in Section 3.7 of the manuscript, we have evaluated various combined options for adaptable HONO source outputs, either by increasing the available HCl (we tried temperature, but multiple permeation devices are also an implicit option; Section 3.7.1), and multiple NaNO2 devices (Section 3.7.2).

This is already stated at Lines 611-612:

'The use of multiple HCl permeation tubes in a single oven, in series, or in parallel are additional options to control the HONO mixing ratio generated in the calibration system.'

Operating parallel calibration channels, as the Reviewer suggests, at 1-2 orders of magnitude higher or lower mixing ratios could be easily accomplished and using a larger Al-block with more holes could increase the versatility further.

We have added this excellent point to our discussion at Lines 723-725:

'The HONO calibration source was designed to facilitate multiple calibrant concentrations, as the four holes in the aluminium heating block (Fig. S3) allows for the operation of parallel HONO sources if desired.'

(RC) Section 2.3 and S2: The set-up of the HCl permeation source is not clear for me. Is the 1/4" tube filled with liquid HCl inserted into a 1/2" PFA tube which is temperature controlled in the Al-block and flushed by the zero air? Is the smaller tube fixed in the middle of the larger PFA tube (how?) or is it simply lying on the ground of the larger tube? Please show a more detailed figure of the HCl permeation source.

(AR) The Reviewer has correctly identified how the HCl permeation source operates. This is the same as any commercial permeation source operates and does not warrant a more detailed figure.

However, for readers who are not familiar with commercial permeation ovens, we have clarified these specific details at Lines 205-207:

"During operation the HCl PD is placed within the $\frac{1}{2}$" tubing in the heating block, through which the carrier gas is flushed."

(RC) Line 209 and Fig.1: Why is the low HONO source flow rate of only 100 sccm used and later diluted to 1.1 slm? If the source should be tuned to high HONO levels (ppms, see line 664), the HONO concentration in these 100 sccm are 11 times higher leading to stronger decomposition of HONO by R3. Why not using 1.1 slm for the reaction?

(AR) The reaction to produce HONO relies on the complete heterogeneous uptake of HCl by the coated NaNO2 reaction device. Increasing the flow will potentially decrease the reaction efficiency. The mixing ratios in our HONO source are lower by a factor of 2000 compared to all other sources presented in the literature (Febo et al., 1995; Gingerysty and Osthoff, 2020; Taira and Kanda, 1990), and is therefore the least susceptible to the decomposition of HONO by R3. The HONO calibration community has widely used percentages of impurities to market the quality of their sources. However, we note that this leads to biased perspective as the absolute quantities of impurities

scale with the amount of HONO produced. Regardless, we have addressed this issue in other responses to the major comments from the Reviewer above.

(RC) Line 214: Typically also 1 % glycerol is added to a coating solution of a denuder (see coating of the NaNO2 reaction devices, see line139)?

(AR) This is explicitly detailed in Section 2.1, which is titled 'Coated NaNO2 Reaction Devices' and specifies our approach. From Lines 145-150:

'A NaNO2 (EMSURE[®]; ACS Reag. Ph Eur, Germany) coating solution was made as a 20 g L-1 NaNO2 solution. The coating solution solvent was composed of equal volumes of methanol (HPLC Grade; Fisher Chemicals, Ottawa, ON) and 18.2 M$\Omega$Åůcm deionised water with 1.0 g L-1 glycerol (Sigma Chemical Company, St. Louis, MO, USA) to facilitate a uniform salt coating. The solution was made by dissolving the NaNO2 in the water first, followed by the addition of the glycerol and then methanol.'

(RC) Section 3.2: caused by the non stable reaction by "H2O" (or CO2?) this side reaction should be minimized, see major concern. I cannot follow the statement in lines 335-337 – HONO production by this reaction is not defined and not constant, see Figure 6!

(AR) We acknowledge that more work needs to be done to understand this potential water-driven mechanism. We do not believe the reaction is caused by the presence of CO2. As stated in our response to the major concern of the Reviewer above, our compressed gases were ultrapure cylinders and our deionised water system minimizes the uptake of CO2. Figure 6 was produced using an NaNO2 coated annular denuder in the presence of HCl and it is not clear why the Reviewer is conflating this Figure, presented 10 pages later in the manuscript, with the results of the water reaction investigated in isolation with water vapour alone in Section 3.2.

We were able to isolate the water effect using the denuders, yet we also stated that these devices had unstable ouput which was not observed with the PFA devices. We

have presented a robust mass balance analysis of the stability of this side reaction within PFA devices in Table 2, although the HONO quantities generated challenged our instrumental detection limits, leading to the denuder-based experiments.

Regardless, at Lines 646-648 we clearly state that the denuders generated very drift-prone HONO outputs when attempting to generate higher HONO mixing ratios:

'The major drawback of using an annular denuder is that the output drifted to lower mixing ratios continuously at a rate of a few ppbv per hour, which was not a feature of the PFA devices (Figure 6).'

(RC) Lines 350-353. For me the source is already unstable at 40 C, see peaks at 2 PM in Figure S7.

(AR) The Reviewer is incorrect that the variance in the output increases at 40 C. The time period they are referring to corresponds to 50 C. At this time off-gassing of HCl from the interior walls and fittings of the permeation oven is occurring. This has been described in the manuscript.

We refer the Reviewer to the following statements from the manuscript at Lines 362-367:

'The HCl output was found to be temperature-dependent and increased exponentially with temperature, as expected from theory (Section S2). However, as the PD was ramped to higher temperatures (> 50 °C) the permeation rate became more unstable, with a resulting settling time of about an hour as the materials from the permeation oven apparatus re-equilibrated (Figure S7). Since the HCl PDs were observed to be most stable at 30 °C and 40 °C, these temperatures were considered optimal to generate the stable HONO mixing ratios.'

To clarify the temperatures being used in Figure S7, we have added additional windows demarking the stable regions in the attached figure:

'Figure S7: Time series of the measured HCl output from PD-6b using CRDS, as

well as temperature of the oven. The blue-shaded bars indicate the region where HCl output was considered stable and this data was used to calculate the variance shown in Figure S6.'

(RC) Table 1 and section S2: The production of the HCl tubes seems to be really "tricky" since similar tubes produce variable HCl emission by one order of magnitude! In addition the given errors of 0.01 ppb for the first three HCl concentrations are only a short term precision but not for longer time, compare data in Tables 2+3.

(AR) Yes, there is an art to sealing the PTFE rods in the end of the permeation tubes that can result in large variability. We have addressed these nuances in the manuscript and in detail to the HCl permeation device stability raised by the Reviewer in their major concerns above. We have also proposed an alternative method to potentially obtain reproducible emission results, in line with commercially available permeation devices, to encourage continued use of economical custom-built permeation devices.

(RC) Line 380-384: Why are the HCl emissions decreasing by one order of magnitude for a time period of 2 years? Is this in accordance with the expected decrease of the liquid HCl concentration (e.g. 6 M) from the loss of gaseous HCl? In addition, a 30 % decrease is not similar to an order of magnitude (2 -> 0.21).

(AR) Permeation device emissions are well known to decrease over time. The actual amount it decreases will depend on use, the quantity of the chemical species, and how well it is made. The 30 % came from another study using permeation devices $\frac{1}{4}$" in diameter compared to ours which are 1/8" and it may be that this particular PD decreased faster as they contain much less solution. We advise regular check on the permeation device outputs, which is standard practice for this equipment.

To clarify this point we have included additional text in the Conclusions:

'Custom-made HCl PDs are prone to variability in emission rates, both between similarly made PD and over time, and therefore require regular characterization, but can

provide a stable output over the order of weeks'.

(RC) Table 2: ADEs of only >55% and >84 % can be determined based on the given errors of HONO (e.g. 0.14/0.31: : :) and not >99%.

(AR) We measured the HCl exiting the reaction devices and could not detect it by our high sensitivity cavity ringdown system (Line 258: DL = 5 pptv; Lines 437-438: 'No HCl was observed to exit the devices, indicating unit acid displacement efficiency and reaching mass balance.') . The error of the HONO measurement is not required to then calculate the >99 % ADE values. Lines 437-438 noted above immediately precede this table and it is not clear how our approach can be clarified further.

(RC) Figure 2: The red shaded area is almost invisible during zero. In addition I do not see any blue shaded area?

(AR) The shaded blue region is rightfully difficult to discern as it is nearly the same width as the line. Numerically, it is 37+/-12 counts, and not above the CIMS detection limits for m/z 35, which we have presented above in response to another comment from the Reviewer.

We have increased the saturation intensity of the red shaded region in the attached Figure 2 to make it easier to see in the zero regions and amended the Figure caption regarding the variance in the HCl trace:

'Figure 2. Conversion efficiency of HCl (blue) to HONO (red) via the acid displacement reaction on a NaNO2 reaction device. The HCl PD-6a and one coated PFA device were used and measured following two hours of stabilization. The acids were observed by acetate quadrupole CIMS with time resolution of 0.50 s and averaged to 60 s. Yellow shaded regions indicate the addition of zero air to the instrument inlet for background correction, while red and blue shaded regions correspond to $1\sigma$ variance in the observations. Note that the variance in the HCl trace is similar to the width of the line.'

(RC) Line 458-459: That is not true for Run1 in Figure 3 (>24 h)

(AR) We have already addressed this misperception in a previous comment from the Reviewer and have revised the figure that was creating confusion.

(RC) Figure 3, 5, S9, S10, S12: Why do the authors not use a more precise HONO instrument or higher HONO levels? The precision of 0.4 ppb of the NOy monitor is too low to determine any dependencies at HONO levels of ca. 2 ppb. Typically the concentrations should be more than 10 times higher than the precision of the instrument used

(AR) We used a standard commercial NOx monitor for two reasons: 1) this is what we had available in the lab for quantification, and 2) we wanted to keep the system to be as low-cost as possible and use instruments that are widely available. Previous work has demonstrated that using a NOx monitor can give reliable measurements of HONO (Febo et al., 1995; McGrath et al., 2019; Zhou et al., 2018) but we do agree that we are limited by the precision of the NOx monitor. However, we note that we did use more precise instruments: BBCEAS, CIMS, and a gold catalyst NOy in this work to already address this limitation explicitly. In all cases, the uncertainty in the measured HONO from the source was equal to the instrument noise, which demonstrates that the uncertainty in the HONO source output was related more to the instrument than to the stability of the source (see first paragraph Section 3.5). We also note that we have applied statistical evaluation of the source output, consistent with the original work of Febo et al. (1995), by averaging the measurements over longer durations to get a more representative picture of the HONO stability.

Our approach is specifically designed to generated atmospherically relevant mixing ratios. It does not translate well to higher HONO levels, because we are limited by the concentration outputs of the permeation devices.

(RC) Line 477 and Figure 4: I do not understand the low stability of the HCl/HONO production (variability by factors, see Figure 4b: : :)? From where are these peaks coming even after 15 days of use, see Figure S10?

(AR) We have addressed this issue above in response to numerous comments from the Reviewer. We point out here that Figure S10 has time units of hours, not days, as the Reviewer states.

(RC) Line 501: should be "after" and not "before" or "Figure 4b-c"? In Fig 5 the output is more or less stable (+-25%...).

(AR) The Reviewer is correct that this should be 'Figure 4b-c' and we have corrected the typo.

(RC) Line 505-506: I cannot follow that statement, see major concerns.

(AR) We have addressed the major concerns and hope that the Reviewer can now follow these statements.

(RC) Line 563-565: Both data shown in Figure S10 are not stable (+- factor of 2) and I cannot follow the RSD of only 16.5%? Cf. the red data at 5, 10 and 37 h.

(AR) We addressed this concern in the responses above and do so again here. We refer the Reviewer to Lines 576-584:

'We next tested the reproducibility for newly made HCl PDs. Two experiments used PD-6b, containing 6 M HCl (Figure S10). After a period of stabilization, the two experiments generated similar HONO mixing ratios (2.58$\pm$0.43 ppbv after 25 hrs, RSD = 16.5 %, RSE = 1.43 %, n = 792). The spikes in HONO output at 15 h and 21 h in the second experiment (green trace, Fig. S10) were likely due to pulses of HCl which we commonly observed with new PDs (e.g. Fig. 4a-b). This emphasizes our recommendation that new custom-HCl PDs should be used for an extended period prior to use for acid displacement to ensure the emission rate has stabilized.'

Stability of the output was measured after 25 hours of operation in both runs, each with a new permeation device. We have already explained in the manuscript the source of the transient emissions of HCl from newly prepared permeation tubes as well.

(RC) Line 592: The relative stability of the HONO source is lower at 40 _C compared to 50_C see Table 3 (1+-0.5 = +-50% and 2+-0.5 = +-25 %). So I do not understand that statement?

(AR) We have rephrased to improve clarity:

'Part of the observed variability in HONO emissions at 50 ËŽC was contributed by the increasingly unstable emissions of HCl at this temperature (e.g. see Fig. S6).'

(RC) Line 606-607 and Table 4: I do not understand the lower output of the quartz tubing? Normally quartz is hydrophilic and can be much better wetted by aqueous solutions than the very hydrophobic PFA, leading to higher expected nitrite levels for the quartz?

(AR) We agree that quartz is normally more hydrophilic than PFA, but we believe that this result was due to the poor coating of nitrite solution as was observed visually when making this device.

We are not sure why and we have adjusted the text at Lines 624-625 to read:

"The quartz tubing gave a notably lower HONO output compared to other materials. This may have been due to a poor coating efficiency on the surface, as observed visually when making this device. This is an unexpected outcome given that quartz is more hydrophilic than PFA."

(RC) Lines 624-630: The reason for the unstable and much higher HONO emissions (Âż[HCl]) is the higher surface area and the unwanted side reaction by "H2O" (or CO2?), see above

(AR) We agree that the higher HONO emissions with the annular denuder were due to the higher surface area and this is already stated in the first sentence of the last paragraph in Section 3.7.2, Lines 642-644:

'More HONO can be generated using the same PDs in conjunction with an annular

denuder, which has a larger internal surface area of 3063 cm2 compared to 388 cm2 for the PFA device.'

(RC) Line 643-644: The source by Febo et al. was much more pure (>99.5%) than the present one (see 6 % NO = 12 % NOx, see R3 => purity 88%...).

(AR) This comment has already been addressed in detail in prior responses. The measurement of impurities at these very low levels is not trivial. For example, it is important to consider relative versus absolute purity of the gases measured when comparing techniques. The Febo source generated tens of ppmv, whereas our source generated a few ppbv of HONO. This means observing impurities in our source is i) more challenging to perform and ii) likely to arise from pathways other than HONO self-reaction (R3). For this reason, we used the highest sensitivity NO instrument developed to date to pursue this task, a pair of world-class NO2 instruments, and compared the observations from those instruments to our extensive NOx analyzer dataset to reach our conclusions. Given that all HONO sources must be validated for their output, the issue of purity was explored more for comparison to the literature rather than to verify the acid displacement assumption (i.e. moles of HCl in equal the moles of HONO out).

(RC) Line 662-664: While the former study showed impurities in the 10 % range at 4 ppm, this source shows similar impurities already below 1 ppb (see line 657), which is very untypical with respect to the quadratic reaction kinetics of reaction R3. And again in Febo et al. the purity was higher than 90% even in the ppm range.

(AR) We do not posit the HONO self-reaction of R3 to be the mechanism by which the observed NOx species are produced at the much lower HONO mixing ratios generated in our source. The mechanism must be something else. See prior responses for expanded details above.

(RC) Line 675: Should be <25 % see line 560?

(AR) The error here was writing RSD, which should have been RSE. This has been

corrected in the manuscript.

(RC) Line 858: Processes

(AR) This suggestion is incorrect. The abbreviated notation we have applied is correct.

(RC) Line 887 and 902 Res. Atmos.

(AR) Corrected.

(RC) Line 988: 10155-10171

(AR) This is a formatting issue in the AMT template for Mendeley. The correction has been entered manually.

(RC) Line 993: 9093-9106

(AR) This is a formatting issue in the AMT template for Mendeley. The correction has been entered manually.

(RC) Line 999: delete the second Vecera, Z.

(AR) Corrected.

(RC) Line 1010: D23302

(AR) Corrected.

(RC) Line 1061: L15820

(AR) Corrected.

(RC) Line 1074: 1681

(AR) Corrected.

(RC) Supplement Section S1, second paragraph, line 5: should be 47 and 51 sccm, see Figure 1?

(AR) We have corrected this to read ∼50 sccm.

(RC) Section S1, last paragraph before Figure S1: The temperature should be electronically limited to 150 _C (see harmful degradation products for PFA in the case of a malfunction).

(AR) We have added a note here regarding this advice:

'An upper limit on the temperature controller should be set at 150 °C to prevent thermal degradation of PFA tubing in the Al-block during use.'

(RC) Figure S3: Why are larger hole diameters (0.6") used in the Al block for the 0.5" PFA permeation source tubes? There would be better heat transfer, if the holes would be only slightly larger than the PFA tubes (e.g. 0.51") getting them into direct contact with the aluminium.

(AR) The 0.5" for the PFA tubes in the permeation oven is their inner diameter and the external diameter is 0.6". They fit very tightly in the machined holes to ensure firm contact for heat transfer.

We have added a statement to clarify this:

'The machined holes are the exact size of the outer diameter of the $\frac{1}{2}$" PFA tubes, resulting in firm contact between the polymer and Al.'

(RC) Figure S7. From where do the peaks during the temperature ramps (40-50_C and 50- 60_C) result? There should be constant increases of HCl similar to the ramp between 30-40 _C?

(AR) See our response to prior comments above. The HCl is adsorbed on and in the PFA polymer, as well as the permeation oven fittings. When the temperature is raised, the HCl desorbs from the surfaces, while the permeation device emissions adjust slowly, resulting in the observed peaks.

Anonymous Referee #2 (RC) Lao et al. present the design of a HONO calibration

source suitable for field use. The manuscript gives a detailed description of the design of the source and extensive information on its performance. The paper solves a challenge that the community currently faces, the lack of reliable HONO calibrations of in-situ instrument. It is thus timely and highly relevant.

Overall the paper is very well written and informative. It provides all the details needed for the community to reproduce the study and, most importantly, build their own HONO calibration source. I did not find any real issues with the manuscript, and my more detailed comments below are just requests for some minor clarifications. Overall, the manuscript is well suited for publication in AMT, and I recommend for publication after a few minor clarifications.

(AR) We thank the Reviewer for their comments and have addressed their requests for clarifications in our responses below and in the main manuscript.

(RC) As water bath operates at 40C producing a RH of 50%, I am wondering if there is an issue with condensation once the air flow cools down while exiting the oven and/or instrument. This may be especially relevant if the system is used in cold climates. Could you please comment on this in the manuscript?

(AR) The impinger/bubbler to control the RH at 50 % operates at room temperature, not 40 °C (see depiction in Figure 1).

We recognize that this is not clearly stated in the description in Section 2.2 and we have clarified the text from Lines 185-189:

'The first critical orifice connects to the HCl PD channel within the heated Al block and the second connects to a 25 mL glass impinger (EMD Millipore Corporation, Billerica, MA, USA) containing deionised water at room temperature. The flows are combined and mixed to a resultant relative humidity (RH) of 50 %, which then enters the coated NaNO2 reaction device in the temperature-controlled Al block.'

We appreciate the Reviewer's point about condensation and have also added text to

caution against condensation at Lines 193-194:

'If operating in cold environments, care should be taken to ensure the 50 % RH exiting the calibration system does not generated condensation in the lines.'

(RC) Could you explain in the manuscript how you determined relative humidity? Was it measured or calculated from the saturation vapor pressure and the gas flows? In the latter case, did you check that the calculated RH is in fact correct?

(AR) We calculated the RH using saturated vapour pressure. We have clarified that it is the saturation vapour pressure and mixing of equal flows in the modified text at Lines 185-189 noted in the prior response by adding 'resultant'.

We have measured this numerous times in the past for producing 50 % RH in HONO sources with loose salt beds, and given the reproducibility of this approach across several years of experimental work (MacInnis et al., 2016; McGrath et al., 2019; Roberts et al., 2010; VandenBoer et al., 2015; Zhou et al., 2018), did not repeat the measurements as part of this work.

(RC) You mention that the system has been found to be stable for approximately one month. Could you clarify if this means that it yielded a constant HONO concentration for this time period, or was there a drift in the concentration due to aging of the NaNO2 reaction device?

Could you comment on whether the length of the tubing between the source and the HONO instrument (or NOx instrument) has an impact?

(AR) We thank the Reviewer for these comments. For the stability, at Lines 539-540 we report a constant HONO mixing ratio of 2.07 +/- 0.48 ppbv for the time period of a month. After 5 weeks, we observed the mixing ratios started to decline from the depletion of NaNO2 in one coated device.

We have added a comment on this at Lines 540-542: 'These measurements used a single NaNO2 reaction device over 5 weeks of continuous operation, after which the

depletion of NaNO2 resulted in a decline of HONO mixing ratios.'

From the range of NaNO2 mass coated on each device, we conservatively estimated that these would be stable for a month, using our observations with many additional coated devices producing stable HONO over periods of 4 weeks as a second justification that this was a sound estimate.

We have clarified this point in the manuscript at Lines 312-315:

'In practice, we observed a PFA device generating approximately 2 ppbv min-1 of HONO to be reliable for over four weeks during experiments performed to test the stability and reproducibility of the PFA devices (Sections 3.5, 3.6).'

We are happy to comment on the impact of tubing length for the Reviewer. In general, adsorption effects in the very clean PFA lines of the calibration source are not an issue and are minimal on glass surfaces as well (VandenBoer et al., 2015). We cannot comment on losses to other tubing materials and HONO losses to inlets sampling ambient air have been noted previously (Zhou et al., 2002). The impact of the length of PFA tubing between the source and instrument depends on the temperature of the lines, residence time of the HONO, and the surface area to volume ratio. With decreasing temperature and increasing SA/V for a given residence time, adsorption losses of HONO to the surface would increase, but eventually equilibrate (likely on the timescale of minutes or less under most relevant operating conditions). Decreasing residence time of the HONO in the lines is the easiest approach to mitigate any potential losses and we suggest using the shortest lines possible leaving the calibration source prior to being diluted into a low residence time flow to deliver to instruments. If the calibration lines can be heated to $> 30\,^{\circ}$C, then line effects can be effectively nullified (VandenBoer et al., 2013).

Anonymous Referee #3 (RC) Lao et al. presented the design and detailed test results of a system capable of producing a wide range of concentrations of gas-phase HONO. The system is a modification to that by Febo et al (1995), based on displacement of

nitrous acid from solid sodium nitrite by gaseous hydrochloric acid. The modifications include the uses of a custom built HCl permeation source and a NaNO2-coated tube (or denuder) for the HONO displacement to take place. These modifications indeed make the system more compact in size and easier to transport. The tests are comprehensive, the results are well presented, and manuscript is well prepared. The manuscript can be helpful for laboratories in constructing a portable HONO generation system for calibrating field HONO instruments and for atmospheric HONO chemistry research in the laboratory. However, I find following major issues with the manuscript:

(RC) 1. Compared to the original design by Febo (1995), the system described in this manuscript offers more compact in size and is capable of generating lower HONO source at concentrations (sub ppb vs 5 ppb). However, there are no improvements in stability and reproducibility (âĹij24% vs <0.4%), and purity (>90% vs 99.5%) of the HONO source generation. The NOx analyzer used to quantify HONO output has a lower detection limit of âĹij0.4 ppb. The poor signal stability of the NOx analyzer near its detection limit may in part be responsible for the not-so-great performance of the system.

(AR) We have previously addressed in detail the stability and purity of our source relative to the design from Febo et al. (1995) in our response to Reviewer 1, major comments 2 and 3.

We agree that we were limited by the precision and detection limit of our NOx analyser, and much of the apparent noise that we see from the calibration source we believe is due to the noise on the instrument used for quantification. The lack of comparability when using percentages is not surprising. Febo et al. (1995) measured their output stability with a NOx analyzer from a source generating dramatically higher HONO mixing ratios (on the order of ppmv). Thus, the relative variance can easily be lower than in our report as the absolute variance from the prior work is many ppbv. In terms of stability, as we note in our manuscript it is difficult to make a solid comparison to the prior work as the statistical handling of the data (i.e. averaging period) and measurement parameters (i.e. measurement rate, use of active filters) were not reported, but the stabilities are comparable using some reasonable assumptions. The stability of our source may even be better and this is demonstrated by the fact that the noise from the calibration source corresponded to the instrument precision of all devices that measured the HONO output rather than actual fluctuations in the output of the calibration source, as stated at Lines 476-481:

'The noise observed in the stabilized HONO output in Figure 3 can be primarily attributed to the noise associated with the NOx analyzer detector (18 of the 24 %; DL =465 0.4 ppbv; 1-minute average). This conclusion is supported by the lower noise in ~2.5 ppbv HONO mixing ratios observed by the CIMS (Fig. 2, RSD of 8.1%), ACES (RSD 8.2 %), and NOyO3 (RSD 1.9 %). In these added observations with higher sensitivity instrumentation, the stability was equal to instrumental precision.'

(RC) 2. The HONO gas stream generated by this system must be calibrated by a primary instrument before it could be used as a HONO calibration source. Therefore, the system itself is not a tunable calibration source for HONO as the authors claimed in the title. A better HONO measurement will be needed to calibration the HONO source at low concentrations; the NOx analyzer described in the manuscript is not adequate for this purpose.

(AR) We were referring to the output of the calibration source as tunable, as the output can be changed through adjustments in oven temperature, different HCl PDs, and dilution flows. Simply put, the tunability of previously published approaches by changing liquid flows or solution concentrations as in Taira and Kanda (1990), or operating additional pumps to reduce the HONO concentrations as in Gingerysty and Osthoff (2020) are cumbersome and expensive compared to simple changes to dilution flows on our source, which produces substantially lower mixing ratios of HONO to reach an environmentally-relevant work range. Gas phase dilutions are abundantly described for HONO in the literature, often being 'tuned' on the order of seconds or, more often, dictated by the instrument response time. If the Reviewer's point is that our calibration

source cannot adjust the output across orders of magnitude in a systematic fashion quickly, that is also incorrect, but could require 1 or 2 hours should such large changes be needed. This is done by changing the temperature of the heating block, swapping permeation devices, or operating multiple channels in parallel. However, we have addressed this apparent need for instantaneous tunability in response to Reviewer 1 by noting that the heating block can simply house multiple parallel calibration channels that cover the necessary range.

The remainder of the calibration work is rapidly achieved by modifying the dilution flow, as stated in the abstract at Lines 22-23: 'Mixing ratios at the tens of pptv level are easily reached with reasonable dilution flows.' Please see the added text in the conclusions to emphasize this point at Lines 723-725: 'The HONO calibration source was designed to facilitate multiple calibrant concentrations, as the four holes in the aluminium heating block (Fig. S3) allows for the operation of parallel HONO sources if desired.' The versatility of multiple permeation devices and parallel channels for tunability are already stated at Lines 611-612:

'The use of multiple HCl permeation tubes in a single oven, in series, or in parallel are additional options to control the HONO mixing ratio generated in the calibration system.'

We have previously addressed in detail the issue of using a commercial NOx analyser to calibrate the output of our HONO source in an earlier response to Reviewer 1. Briefly, we choose to use a commercial NOx analyser as this was available and also we wanted to keep the total system as low-cost as possible.

(RC) 3. A commercial HONO source is available, based on the reaction of NaNO2 with diluted H2SO4 solution in a stripping coil (Taira and Kanda, 1990; Kleffmannet al., 2004) (https://qumashop.de/images/LOPAP%2003%20HONO%20Source%20short_v2.pdf). A unit was used during the FIONA intercomparison in 2010 in Valencia, Spain, to

generate a HONO source being distributed to and shared by the collaborating groups. Based on my own experience from this intercomparison, the system appeared to perform significantly better than the system described in the manuscript. Specifically, it offered high precision and stability (âĹij 1% at low-ppb concentrations) and is truly tunable (stabilized within minutes after switching to a different concentration). Of course, the commercial unit could be much pricier than the home-build system described in this manuscript.

(AR) As we stated in our response to Reviewer 1, the unit used in the FIONA intercomparison has not been described in the peer-reviewed literature and so we are unable to compare our source to the results mentioned by Reviewer 3. We also note that low ppbv concentrations are not sub-ppbv mixing ratios, which are typically observed in the daytime troposphere. This means that the modified source used during FIONA was not readily reaching environmentally-relevant mixing ratios, particularly near the daytime minima of 10-100 pptv of HONO. Our source can reach these mixing ratios with simple gas phase dilutions and this means it is far more easily tuned in this range. We have added more text to clarify the lower purity of our source relative to previous work, please see our response to Reviewer 1, major comment 4.

As the Reviewer rightly points out, the HONO calibration source presented in the current work is likely cheaper than the commercially available system mentioned from QUMA, as one of our project goals was to keep the system as low-cost as possible.

[Figure]

[Figure]

**Fig. 1.** Revised Figure 3 - see revised caption in response above

[Figure]

**Fig. 2.** Revised Figure S7 - see revised caption in response above

[Figure]

**Fig. 3.** Revised Figure 2 - see revised caption in response above